# Ensemble cryoEM elucidates the mechanism of insulin capture and degradation by human insulin degrading enzyme

Zhening Zhang[1†], Wenguang G Liang[2†], Lucas J Bailey[3], Yong Zi Tan[1,4], Hui Wei[1], Andrew Wang[2], Mara Farcasanu[2], Virgil A Woods[5], Lauren A McCord[2], David Lee[5], Weifeng Shang[6], Rebecca Deprez-Poulain[7], Benoit Deprez[7], David R Liu[8], Akiko Koide[9,10,11], Shohei Koide[9,10,11], Anthony A Kossiakoff[3], Sheng Li[5*], Bridget Carragher[1,4*], Clinton S Potter[1,4*], Wei-Jen Tang[2*]

[1]National Resource for Automated Molecular Microscopy, Simons Electron Microscopy Center, New York Structural Biology Center, New York, United States; [2]Ben-May Institute for Cancer Research, The University of Chicago, Chicago, United States; [3]Department of Biochemistry and Molecular Biology, The University of Chicago, Chicago, United States; [4]Department of Biochemistry and Molecular Biophysics, Columbia University, New York, United States; [5]Department of Medicine, University of California, San Diego, La Jolla, United States; [6]BioCAT, Argonne National Laboratory, Illinois, United States; [7]Univ. Lille, INSERM, Institut Pasteur de Lille, Lille, France; [8]Department of Chemistry and Chemical Biology, Harvard University, Cambridge, United States; [9]Perlmutter Cancer Center, New York University School of Medicine, New York, United States; [10]New York University Langone Medical Center, New York University School of Medicine, New York, United States; [11]Department of Biochemistry and Molecular Pharmacology, New York University School of Medicine, New York, United States

**\*For correspondence:**
Corresponding authorss4li@ucsd.edu (SL);
bcarr@nysbc.org (BC);
cpotter@nysbc.org (CSP);
wtang@bsd.uchicago.edu (W-JT)

[†]These authors contributed equally to this work

**Competing interests:** The authors declare that no competing interests exist.

**Abstract** Insulin degrading enzyme (IDE) plays key roles in degrading peptides vital in type two diabetes, Alzheimer's, inflammation, and other human diseases. However, the process through which IDE recognizes peptides that tend to form amyloid fibrils remained unsolved. We used cryoEM to understand both the apo- and insulin-bound dimeric IDE states, revealing that IDE displays a large opening between the homologous ~55 kDa N- and C-terminal halves to allow selective substrate capture based on size and charge complementarity. We also used cryoEM, X-ray crystallography, SAXS, and HDX-MS to elucidate the molecular basis of how amyloidogenic peptides stabilize the disordered IDE catalytic cleft, thereby inducing selective degradation by substrate-assisted catalysis. Furthermore, our insulin-bound IDE structures explain how IDE processively degrades insulin by stochastically cutting either chain without breaking disulfide bonds. Together, our studies provide a mechanism for how IDE selectively degrades amyloidogenic peptides and offers structural insights for developing IDE-based therapies.
DOI: https://doi.org/10.7554/eLife.33572.001

## Introduction

Insulin degrading enzyme (IDE) is an evolutionarily conserved, M16 family metalloprotease that controls diverse biological functions in model organisms such as mating and cell division in budding

yeast and growth in fruit flies (*Adames et al., 1995*; *Fujita et al., 1994*; *Galagovsky et al., 2014*; *Tang, 2016*). IDE is ubiquitously expressed in all tissues and can be found in almost all subcellular compartments, despite being made as a cytosolic protein (*Tang, 2016*; *Tundo et al., 2017*). IDE can effectively degrade insulin, amylin, and glucagon, pancreatic hormones that control blood glucose levels, as well as amyloid β (Aβ), a peptide implicated in Alzheimer's disease (*Duckworth et al., 1998*; *Kurochkin et al., 2018*; *Pivovarova et al., 2016*; *Tang, 2016*; *Tundo et al., 2017*). Defects in IDE alter the progression of type two diabetes mellitus and Alzheimer's disease in rodents and are linked to these diseases in humans (*Farris et al., 2003*; *Farris et al., 2004*; *Fuchsberger et al., 2016*; *Pivovarova et al., 2016*; *Tang, 2016*; *Tundo et al., 2017*). Two IDE-specific inhibitors improve glucose tolerance (*Durham et al., 2015*; *Maianti et al., 2014*) and IDE overexpression reduces Aβ load in mice (*Leissring et al., 2003*), making this enzyme a promising therapeutic target (*Kurochkin et al., 2018*; *Pivovarova et al., 2016*; *Tang, 2016*; *Tundo et al., 2017*).

Crystallographic and biochemical studies have provided the framework for IDE substrate recognition (*Guo et al., 2010*; *Malito et al., 2008a*; *Malito et al., 2008b2008*; *Manolopoulou et al., 2009*; *Noinaj et al., 2011*; *Ren et al., 2010*; *Shen et al., 2006*; *Tang, 2016*). IDE is a 110 kDa zinc metallo-protease that readily dimerizes in solution (Kd =~ 10 nM) (*Li et al., 2006*). IDE has at least two major conformational states in its catalytic cycle; open-state IDE captures substrates and releases products while closed-state IDE performs catalysis (*Figure 1A*). Thus far, all crystal structures of dimeric IDE are in the closed state. Within the IDE dimer, each IDE subunit consists of ~55 kDa N- and C-terminal domains, IDE-N and IDE-C (*Figure 1B*). Together, these domains form an enclosed, sizable catalytic chamber, the size of which explains why IDE prefers to degrade peptides that are less than 80 amino acid long (*Figure 1B*). Various substrate-bound IDE structures reveal that the high selectivity of IDE is partly achieved by the specific interactions between the IDE catalytic chamber and substrate via size and charge complementarity (*Guo et al., 2010*; *Malito et al., 2008b2008*; *Manolopoulou et al., 2009*; *Ralat et al., 2011*; *Ren et al., 2010*; *Shen et al., 2006*). The formation of cross-β-sheet between an exposed β-strand in the unfolded substrate and the catalytic cleft in conjunction with the anchoring of the substrate's N-terminus of substrate to a site that is ~30 Å away from the catalytic zinc ion explains how IDE selectively cleaves the exposed β-strand distal to the N-terminus of the targeted peptides in a stochastic manner (*Shen et al., 2006*).

Until now, structures of open and insulin-bound state IDE, two key conformations vital for the IDE catalytic cycle, have remained unsolved (*Figure 1A and C*). In closed-state IDE, substrate cannot enter into the catalytic chamber of IDE and the cleaved products cannot exit. Thus, IDE needs to undergo a significant open-closed transition during its catalytic cycle. In addition to contributing to an understanding of how IDE captures its substrate and releases its reaction products, the structure of open-state IDE can provide the insight into how the open-closed transition of IDE facilitates the unfolding of its substrates prior to the cleavage reaction as well as how IDE conducts its non-proteolytic roles, for example, regulating proteasome activity and preventing amyloid fibril formation of α-synuclein (*Sharma et al., 2015*; *Tang, 2016*). The IDE-insulin interaction represents a unique challenge of how IDE interacts with its substrates. Insulin consists of A and B chains that are held together by two inter-molecular disulfide bonds. IDE processively degrades insulin into two pieces without breaking these disulfide bonds (*Figure 1A*) (*Manolopoulou et al., 2009*). The previously reported crystal structure of insulin-bound IDE reveals how insulin is partially unfolded inside the catalytic chamber of IDE, the first step in the unfolding and degradation of insulin by IDE (*Manolopoulou et al., 2009*) (*Figure 1C*). However, the structure of IDE in complex with the fully unfolded insulin prior to the processive cleavage of insulin has remained unsolved.

Amyloidogenic peptides such as Aβ can form highly toxic oligomers/fibrils, leading to many human disorders (*Chiti and Dobson, 2006*; *Eisenberg and Jucker, 2012*; *Merlini and Bellotti, 2003*). A salient feature of many amyloidogenic peptides is their high propensity to unfold, resulting in exposed β-strands that together form cross-β-sheets and then amyloid fibrils (*Eisenberg and Jucker, 2012*; *Fitzpatrick et al., 2017*; *Lu et al., 2013*). Nucleation to form low-molecular-weight oligomers is a key rate limiting step in the formation of amyloid fibrils (*Merlini and Bellotti, 2003*). IDE selectively degrades certain amyloidogenic peptides, preventing amyloid fibril formation (*Kurochkin, 2001*; *Malito et al., 2008a*). IDE achieves this by cutting only the monomeric form of these peptides and cleaving at sites located at the β-strand vital for cross-β-sheet formation (*Malito et al., 2008a*; *Tang, 2016*). Based on the absence of electron density for the catalytic zinc ion-containing IDE door subdomain in a previously reported Fab1-bound IDE crystal structure, we have put forth a

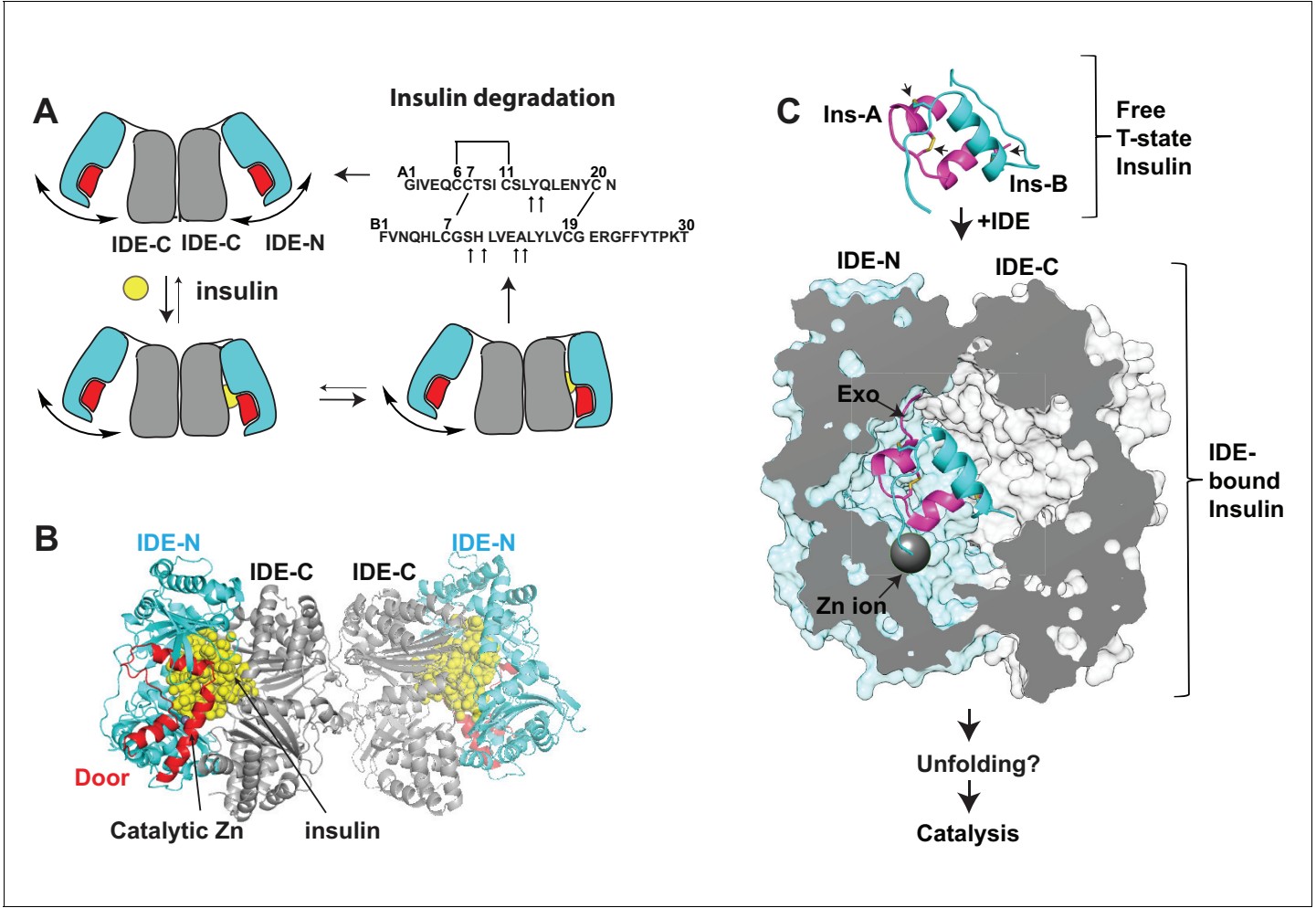

**Figure 1.** Summary of key unsolved IDE structures. (**A**) Schematic diagram for the catalytic cycle of IDE. IDE is dimerized via the interaction of IDE-C domains. The open-closed transition of IDE dimer is postulated to be mediated by the rigid-body motion between IDE-N and IDE-C. The binding of substrate, e.g., insulin, shifts the equilibrium from favoring the open-state IDE to predominantly closed-state IDE. Until now, the structure of open-state IDE has not been determined. (**B**) Ribbon representation of IDE dimer (PDB code: 2WBY). IDE-N and IDE-C are colored in cyan and grey, respectively. The IDE door subdomain that contains the catalytic zinc ion is highlighted in red and the partially unfolded insulin entrapped inside the catalytic chamber of IDE is colored in yellow. (**C**) The key conformational switches required for insulin unfolding and degradation by IDE. Insulin A and B chains in ribbon representation are colored in magenta and cyan, respectively. Top: Insulin alone structure where the intra- and inter-molecular disulfide bonds are marked by arrows. Middle: IDE-bound insulin structure that contains partially unfolded insulin (PDB code: 2WBY). The N-terminus binding exosite (Exo) and catalytic zinc ion are marked by arrows. Until now, the structure of IDE in complex with the fully unfolded or cleaved insulin is not determined.
DOI: https://doi.org/10.7554/eLife.33572.002

hypothesis that IDE uses substrate-assisted catalysis to recognize amyloidogenic peptides (**McCord et al., 2013**). Upon substrate capture, the open-closed transition of IDE causes amyloidogenic peptides to unfold. The resulting exposed β-strand then binds and stabilizes the catalytic cleft within the IDE door subdomain, leading to stochastic cleavage of these peptides. This hypothesis could explain how IDE uses substrate-induced stabilization of the IDE catalytic site to selectively degrade amyloidogenic peptides. However, this hypothesis has not been formally tested. Thus, the molecular basis for the recognition of amyloidogenic peptides by IDE remains unsolved.

Recent advances in cryogenic electron microscopy (cryoEM) have profoundly transformed structural biology, making it possible to obtain near atomic resolution 3D structures that are otherwise difficult to achieve by crystallography (**Henderson, 2015**; **Merk et al., 2016**). Here, we report cryoEM structures of apo and unfolded insulin-bound human IDE, which is recalcitrant to crystallography. Hydrogen-deuterium exchange coupled with mass spectrometry (HDX-MS) and small angle

X-ray scattering (SAXS) have recently become widely used techniques to characterize the high-order structure of proteins and protein complexes under physiological conditions in solution, nicely complement high-resolution techniques such as crystallography and cryoEM (*Blanchet and Svergun, 2013*; *Marciano et al., 2014*; *Putnam et al., 2007*; *Ward et al., 2013*; *Wei et al., 2014*). We integrate all four aforementioned structure methods to elucidate the molecular basis of how IDE captures, unfolds, and degrades its substrates and how IDE recognizes amyloidogenic peptides.

## Results

### Fab-assisted cryoEM of human IDE

Because IDE in the open conformation has proved recalcitrant to crystallization, we explored the use of cryoEM to study these structures. While IDE readily dimerizes ($K_d$ = 10 nM), the dimer's overall size (220 kDa), pseudo two-fold symmetry and conformational heterogeneity made it challenging for cryoEM (data not shown). We addressed these issues by using Fab-assisted cryoEM (*Kim et al., 2015*; *Lyumkis et al., 2013*; *Wu et al., 2012*). To identify Fabs that bind IDE tightly, we screened a phage-display synthetic Fab library constructed using 'restricted chemical diversity' where positions randomized within the complementarity determining regions are biased toward amino acids enriched in antibody paratopes (*Miller et al., 2012*). This allowed rapid identification of eighteen high-affinity IDE binding Fabs without immunization. $Fab_{H11}$ was chosen because it bound IDE tightly (~1 nM) and only slightly increased IDE activity (*Figure 2—figure supplement 1A–C*). We then rigidified the elbow region between the heavy and light chain of $Fab_{H11}$, which has improved the resolution of several structures that used Fab as the crystallization chaperone (*Bailey et al., 2018*). The resulting molecule, $Fab_{H11-E}$, was then used to determine an X-ray structure of $Fab_{H11-E}$-bound IDE at 3.8 Å resolution (R/R$_{free}$ = 22/27%, *Supplementary file 1*). This structure reveals the binding epitope of this Fab to IDE (*Figure 2A and B*). The tight binding between IDE and $Fab_{H11-E}$ is mediated by a network of hydrogen bonds and van der Waal contacts (*Figure 2B*). The crystal structure revealed a closed-state IDE dimer that is nearly identical to those reported previously (*Figure 2—figure supplement 1D*)(*Guo et al., 2010*; *Malito et al., 2008b2008*; *Manolopoulou et al., 2009*; *Ren et al., 2010*; *Shen et al., 2006*). All crystal structures of IDE dimer solved so far are in the closed state, likely due to the constraints imposed by the crystal lattice. This illustrates the challenge in using crystallography to decipher the structure of open-state IDE.

In order to ensure that Fab binding does not alter the conformation and function of IDE, we applied HDX-MS to assess whether $Fab_{H11-E}$ causes noticeable global change in IDE conformations in solution because HDX-MS not only probes the binding regions of a given protein with its partners but also examines protein dynamics and regional stability (*Chung et al., 2011*; *Li et al., 2011*; *Marciano et al., 2014*). As predicted from the $Fab_{H11-E}$-bound IDE crystal structure, we found epitope residues 374–394 to be the primary region that displays strong reduction in HDX. Two additional regions, residues 297 to 303 and residues 501 to 508, also show moderate reduction in HDX in response to $Fab_{H11-E}$ binding (*Figure 2C*, *Figure 2—figure supplements 2–4*). Together, these three regions form a conformational binding epitope, which is in an excellent agreement with our crystal structure. As there is no major change in HDX between IDE alone and $Fab_{H11-E}$-bound IDE that is distal to $Fab_{H11-E}$ binding site, our HDX-MS data also suggest that $Fab_{H11-E}$ binding has a minimal effect on the conformation dynamics or regional stability of IDE in solution. We only found a minor reduction in HDX in a small region, residues 169 to 198, which is a part of IDE door subdomain that contains the catalytic zinc ion. Interestingly, the stabilization of IDE door subdomain is a key feature induced by substrate binding, which is discussed in details in *'substrate-induced conformational change of IDE'*. This could explain a 50% enhancement of IDE catalytic activity upon $Fab_{H11-E}$ binding (*Figure 2—figure supplement 1C*). Altogether, our data suggests that $Fab_{H11-E}$ does not significantly alter the global structure or activity of IDE. Thus, the cryoEM structures of $Fab_{H11-E}$-bound IDE is likely to represent the conformations of IDE dimer in solution.

We then performed cryoEM structural analysis of the $Fab_{H11-E}$-bound IDE dimer in the presence and absence of insulin. 3D maps were reconstructed from 388,643 and 762,283 particles for apo- and insulin-bound IDE-$Fab_{H11-E}$ complex, respectively (*Figure 2D–F*; *Figure 2—figure supplements 5–10*; *Supplementary file 2–3*). A cryoEM map of the insulin-bound IDE-$Fab_{H11-E}$ was constructed using 218,162 particles initially, refined, and solved at 4.1 Å resolution, which confirms how $Fab_{H11-E}$

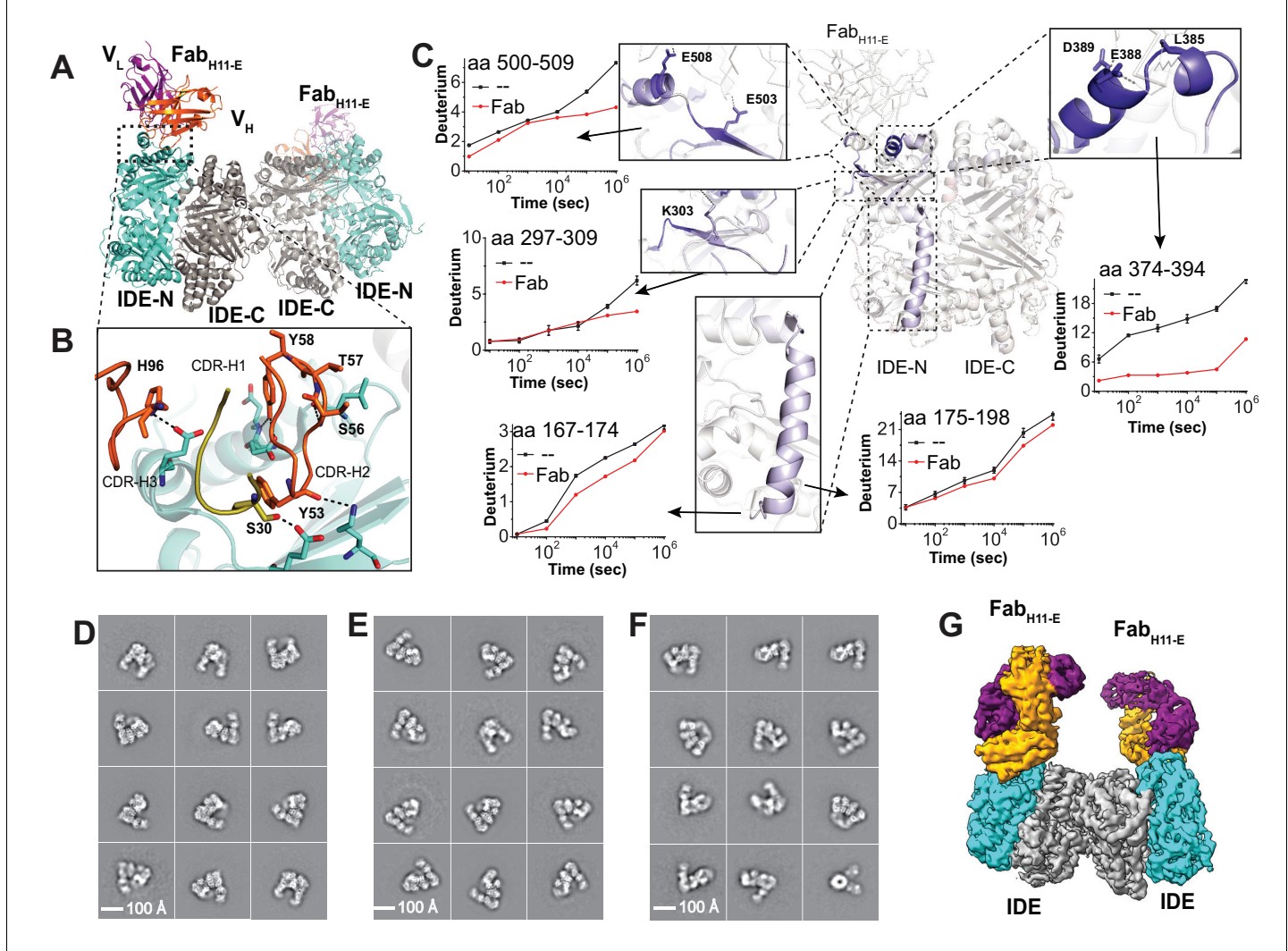

**Figure 2.** Structures of IDE-Fab(H11)-bound IDE dimer. (**A**) Overall structure of Fab$_{H11-E}$-bound IDE dimer in ribbon representation (PDB code: 5UOE). IDE-N and IDE-C are colored in cyan and grey, respectively; The heavy and light chains of Fab$_{H11-E}$ are colored in orange and purple, respectively. (**B**) Detailed interactions between IDE and Fab$_{H11-E}$. The key residues involved in the interaction of IDE with Fab$_{H11}$ were calculated using PDBePISA (*Krissinel and Henrick, 2007*). (**C**) Changes of hydrogen/deuterium exchange of IDE induced by the binding of Fab$_{H11-E}$. Representative 2D class averages of (**D**) insulin-bound IDE-Fab$_{H11-E}$, (**E**) Apo IDE-Fab$_{H11-E}$-bound IDE from untilted micrographs and (**F**) Apo IDE-Fab$_{H11-E}$-bound IDE from 30 degree titled micrographs. (**G**) Electron density map of IDE-Fab$_{H11-E}$-bound IDE dimer in the presence of insulin.

DOI: https://doi.org/10.7554/eLife.33572.003

The following figure supplements are available for figure 2:

**Figure supplement 1.** Characterization of IDE-Fab(H11).
DOI: https://doi.org/10.7554/eLife.33572.004

**Figure supplement 2.** Pepsin digestion maps of IDE for HDX-MS analysis.
DOI: https://doi.org/10.7554/eLife.33572.005

**Figure supplement 3.** Amide hydrogen-deuterium exchange profiles of IDE alone (**A**) and IDE in the presence of Fab-$_{H11}$ (**B**).
DOI: https://doi.org/10.7554/eLife.33572.006

**Figure supplement 4.** HDX-MS analysis of IDE-Fab$_{H11}$ interaction.
DOI: https://doi.org/10.7554/eLife.33572.007

**Figure supplement 5.** Cryo-electron micrographs and 2D class averages of insulin-bound IDE-Fab$_{H11-E}$.
DOI: https://doi.org/10.7554/eLife.33572.008

**Figure supplement 6.** Cryo-electron micrographs and 2D class averages of untilted Apo IDE-Fab$_{H11-E}$.
DOI: https://doi.org/10.7554/eLife.33572.009

**Figure supplement 7.** Cryo-electron micrographs and 2D class averages of tilted Apo IDE-Fab$_{H11-E}$.

*Figure 2 continued on next page*

*Figure 2 continued*

DOI: https://doi.org/10.7554/eLife.33572.010

**Figure supplement 8.** Flow chart of cryoEM data processing.

DOI: https://doi.org/10.7554/eLife.33572.011

**Figure supplement 9.** CryoEM statistics of apo IDE-Fab$_{H11-E}$ and insulin-bound IDE-Fab$_{H11-E}$.

DOI: https://doi.org/10.7554/eLife.33572.012

**Figure supplement 10.** Overall fit of cryoEM density map to the structure models.

DOI: https://doi.org/10.7554/eLife.33572.013

binds IDE (*Figure 2G*, *Figure 2—figure supplement 8B*, *Supplementary file 3*). We further improved the resolution of the IDE dimer using IDE dimer-focused classification with signal subtraction and refinement, resulting in a map and structural model with an overall resolution of 3.7 Å (*Figure 2—figure supplements 8B*, *9* and *10*, *Supplementary file 3*). Apo Fab$_{H11-E}$-bound IDE showed a highly preferred orientation in vitreous ice. Thus, images were acquired with the grid tilted at an angle to the electron beam, which allowed better sampling of other orientations (*Figure 2F*, *Figure 2—figure supplement 9B*) (*Tan et al., 2017*). The initial 3D refinement of apo IDE-Fab$_{H11-E}$ of 151,868 particles had a resolution of 4.4 Å. Further 3D classification revealed three major conformations and structural models were built (*Supplementary file 3*). We again applied IDE dimer-focused classification with signal subtraction of the IDE-Fab$_{H11-E}$ dimer to refine each conformation separately, which improved the resolution significantly (*Figure 2—figure supplement 9A*, *Supplementary file 3*). The three apo-IDE structures derive from combinations of the IDE partial open (pO) and open (O) states (*Figure 3A*). The dimer with one open and one partially open conformational subunit (open/partial open, 110,499 particles) reached an overall resolution of 4.2 Å (*Figure 3A*, *Figure 2—figure supplements 5–9*, *Supplementary file 3*). The dimer with two open conformational subunits (open/open, 24,425 particles) resulted in an overall resolution of 6.5 Å (*Figure 3A*, *Figure 2—figure supplements 5–9*, *Supplementary file 3*). The dimer with two partially open conformational subunits (partial open/partial open, 16,944 particles) reached 6.9 Å resolution (*Figure 3A*, *Figure 2—figure supplements 5–9*, *Supplementary file 3*).

## CryoEM structures of IDE dimer

CryoEM analysis reveals four novel IDE dimer structures, one from insulin-bound IDE and three from apo-IDE (*Figure 3A*). The cryoEM structure of the insulin-bound IDE dimer at 3.7 Å resolution (*Videos 1–2*) shows that both IDE subunits adopt a similar but not identical partially closed (pC) state that differs from the previously reported closed-state IDE (*Figure 3B*, *Figure 3—figure supplement 1*, *Supplementary file 4*) (*Guo et al., 2010*; *Malito et al., 2008b2008*; *Manolopoulou et al., 2009*; *Ren et al., 2010*; *Shen et al., 2006*). The buried surface area between IDE-N and IDE-C and distance between center of mass (COM) of D1 and D4 in the cryoEM pC state is nearly identical to those in the closed-state IDE shown by X-ray crystallographic studies (*Figure 3B*, *Supplementary file 4*). However, this pC state has an approximately 2° decreased dihedral angle between the COM of four homologous domains, D1-D4, compared to those in the closed-state IDE (*Figure 3B*, *Supplementary file 4*), suggesting a potential gliding motion allows IDE to shift between partially closed and closed states. The position of insulin in the catalytic cleft of the cryoEM maps reveals that the IDE pC state is ready for catalysis. Thus, both IDE pC and C states are catalytically competent.

The combinations of the IDE partial open (pO) and open (O) states results in three apo-IDE structures (*Figure 3A*). The IDE pO and O states differs from the pC and C states by 5 Å to 24 Å increases in the distance between D1 and D4 and 7° to 17° increases in the dihedral angle, respectively (*Figure 3B*, *Figure 3—figure supplement 1*, *Supplementary file 4*, *Video 1*). These changes result in decreased buried surface between IDE-N and IDE-C in pO and O states. The three conformers have resolution limits which correlated well with their relative populations: O/pO (4.2 Å, 73%), O/O (6.5 Å, 16%), and pO/pO (6.9 Å, 11%). Interestingly, the conformation of pO and O states in the pO/O IDE dimer differ significantly from those in pO/pO or O/O IDE dimer. Furthermore, two subunits within O/O or pO/pO states have noticeable differences in the distance and buried surface between IDE-N and IDE-C and the dihedral angle (*Supplementary file 4*). Such differences provide a potential explanation for the mechanism by which dimerization-induced allostery regulates the

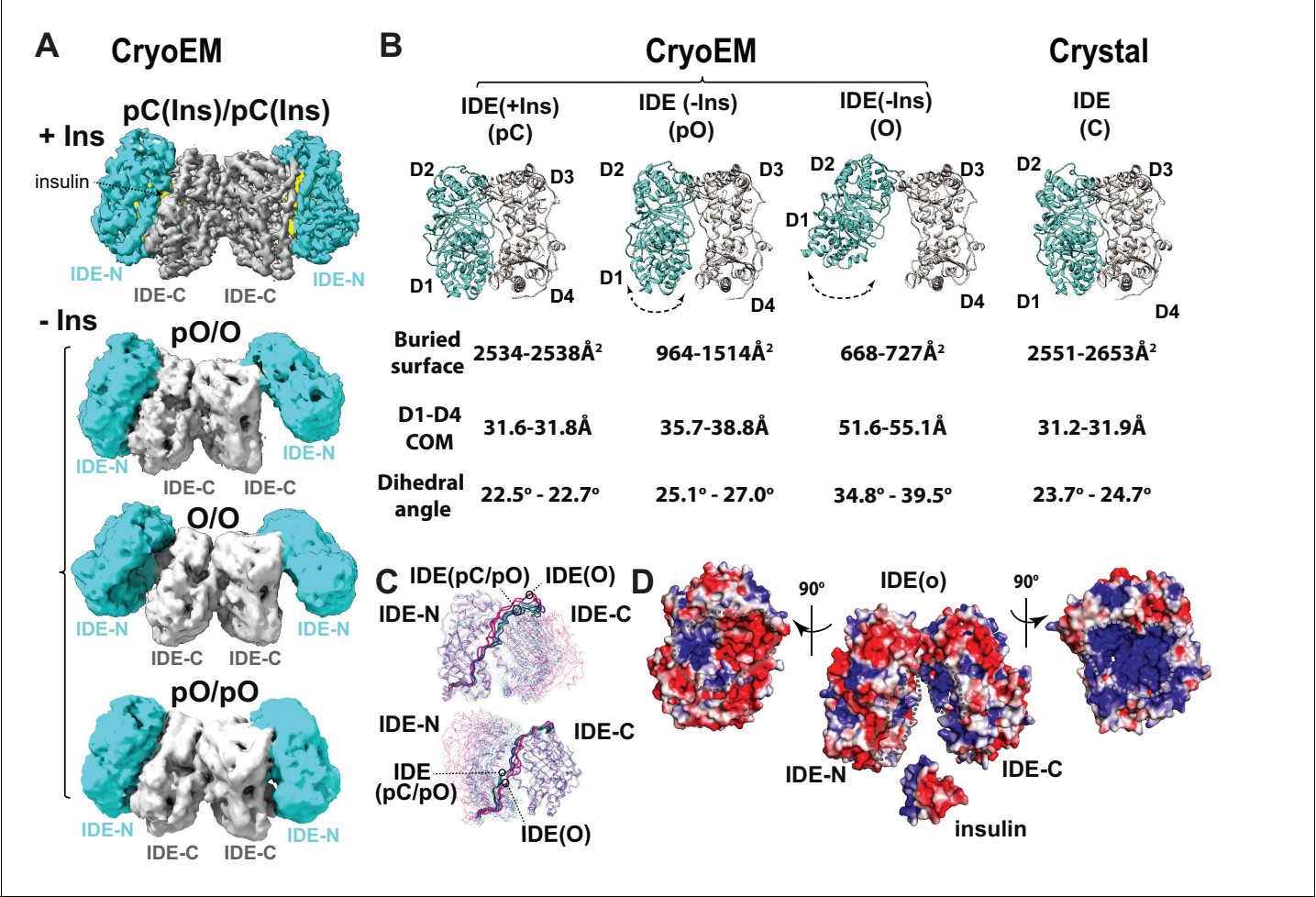

**Figure 3.** CryoEM IDE dimer. (**A**) Overall structures of IDE dimer in the presence and absence of insulin. Models are shown as ribbons within the electron density map. O, open state; pO, partially open state; pC, partially-closed state. IDE-N and IDE-C are colored in cyan and grey, respectively; insulin inside the catalytic chamber of IDE is colored in yellow. For simplicity, Fab is not shown. (**B**) Comparison of four distinct conformational states within IDE structures; two pC states in the insulin-bound cryoEM IDE structure, three pO and three O states in apo-IDE cryoEM structures, and a closed structure from previous crystallographic work (*Guo et al., 2010*; *Malito et al., 2008b2008*; *Manolopoulou et al., 2009*; *Noinaj et al., 2011*; *Ren et al., 2010*; *Shen et al., 2006*). The buried surface between IDE-N and IDE-C, distance between the center of mass (COM) of IDE D1 and D4 domains, and dihedral angles (absolute values) between COM of IDE D1-D2 and COM of D3-D4 are shown below the ribbon presentation of IDE structures. (**C**) Structural comparison of IDE states aligned by IDE-N (top) or IDE-C (bottom), showing rigid body motion of IDE-N and IDE-C guided by the loop connecting IDE-N and IDE-C. (**D**) Structural basis of IDE open structure primed to capture insulin by size and charge complementarity. The boundaries for the substrate-binding catalytic chambers of IDE-N and IDE-C are marked by a dashed line. The color scale is set from −3 kT/e (red) to 3 kT/e (blue) calculated using APBS 2.1.

DOI: https://doi.org/10.7554/eLife.33572.014

The following figure supplement is available for figure 3:

**Figure supplement 1.** Comparison of four distinct conformers of IDE, Open (O), partially open (pO), partially closed (pC), and closed (C).

DOI: https://doi.org/10.7554/eLife.33572.015

catalytic activity of IDE (*Figure 3B*, *Figure 3—figure supplement 1*) (*McCord et al., 2013*; *Ralat et al., 2011*; *Song et al., 2010*). Open-state IDE has an opening just wide enough to capture its substrates, for example, insulin, TGF-α, and MIP-1α/β, allowing the IDE catalytic chamber to attract these substrates with high dipole moment via charge complementarity (*Figure 3D*) (*Guo et al., 2010*; *Manolopoulou et al., 2009*; *Ren et al., 2010*).

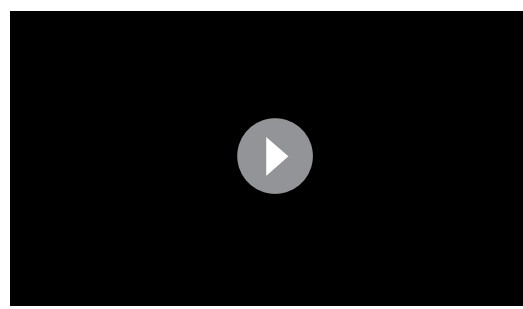

**Video 1.** Overall fit of human dimeric IDE structures with the electron density maps. Scheme 1: CryoEM structure of human insulin-bound dimeric IDE at 3.8 angstrom resolution. Electron density map colored in grey, IDE colored in cyan and in the ribbon representation, and insulin colored in yellow. Scheme 2: Cryo EM structure of human apo-IDE at 4.1 angstrom resolution that has one subunit in the partially-open state and the other in the open state. Electron density map in grey and IDE colored in pale green and in ribbon representation. This is the dominant conformational state of apo-IDE. Scheme 3: Cryo EM structure of human apo-IDE at 5.5 angstrom resolution that has both subunits in the open state. Electron density map in grey and IDE colored in magenta and in ribbon representation. Scheme 4: Cryo EM structure of human apo-IDE at 5.9 angstrom resolution that has both subunits in the partially-open state. Electron density map in grey and IDE colored in purple and in ribbon representation.

DOI: https://doi.org/10.7554/eLife.33572.016

## SAXS analysis of IDE

Our cryoEM structures reveal that only open-state IDE can capture its prototypical substrate, insulin, and release cleaved products. The interaction of open-state IDE with insulin likely facilitates the open-closed transition of IDE. We chose SAXS to test this hypothesis because SAXS provides the ensemble information regarding size and shape of molecules in solution (*Blanchet and Svergun, 2013*). To exclude IDE monomer and larger aggregates, we used size exclusion chromatography in-line with SAXS. Such a SAXS profile of the IDE dimer allows us to assess the distribution of open- and closed-state IDE in the presence and absence of insulin with better precision. We evaluated the experimental data with the prediction from our cryoEM models using the radius of gyration ($R_g$), the average of square center of mass distances of the molecule, and the distance distribution function, p(r) (*Figure 4A*, *Figure 4—figure supplement 1*). Our SAXS data showed that IDE in solution exist in equilibrium between open and partially open states and that insulin constrained the IDE dimer into mostly partially closed or closed states. This agrees with our observation in cryoEM.

Enzyme kinetic analysis estimated that insulin degradation by IDE occurs reasonably rapidly, up to ~2 per second (*Manolopoulou et al., 2009*). To assess whether the insulin-induced open-closed transition of IDE could limit insulin degradation by IDE, we used time-resolved SAXS analysis to obtain rate constants for the insulin-induced open-closed transition of IDE. Time-resolved SAXS experiments were done using a microfluidic laminar flow mixer adapted from the design initially developed for time resolved fluorescence studies (*Park et al., 2008*; *Park et al., 2006*). We found that the change in $R_g$ value caused by the rapid mixing of IDE with insulin fit well with a single exponential decay with τ = 0.1 s (*Figure 4B*). Thus, the timeframe required for insulin to induce conformational switching from a high $R_g$ state to a low $R_g$ state is close to the rate of degradation of insulin by IDE (~2 s$^{-1}$). This suggests that insulin-induced IDE open-closed transition is likely a key rate-limiting step for insulin clearance by IDE.

## Substrate-induced conformational changes of IDE

In a previous crystal structure of the Fab1-bound closed apo-IDE, one subunit within the IDE dimer did not have the electron density for the IDE door subdomain (*McCord et al., 2013*). This leads to the hypothesis that IDE has a catalytic zinc-containing door subdomain that is partially unfolded and/or undergoes a rigid body motion. This hypothesis predicts that, under the crystallization conditions used for Fab1-bound closed apo-IDE crystal structure, the presence of IDE substrates would stabilize IDE door domain, rendering it visible. We thus solved crystal structures for Fab1-bound IDE structures in the presence of Aβ and insulin at 3.5 Å and 3.9 Å resolution, respectively (R/R$_{free}$ = 23/27%, R/free = 24/29%) and Fab1$_{E-}$ and insulin-bound IDE at 3.3 Å resolution (R/R$_{free}$ = 20/25%) (*Supplementary file 1*). Instead of being absent, as in the apo-IDE crystal structure (*McCord et al., 2013*), we found that the door subdomain of IDE is clearly visible in these structures (*Figure 5A*). In fact, these IDE structures are nearly identical to closed-state IDE determined in the previously reported insulin- or Aβ-bound IDE (RMSD = 0.46 and 0.56 Å, respectively, *Figure 5—figure supplement 1*). Consistent with that notion that this is induced by substrate, unfolded Aβ or insulin is

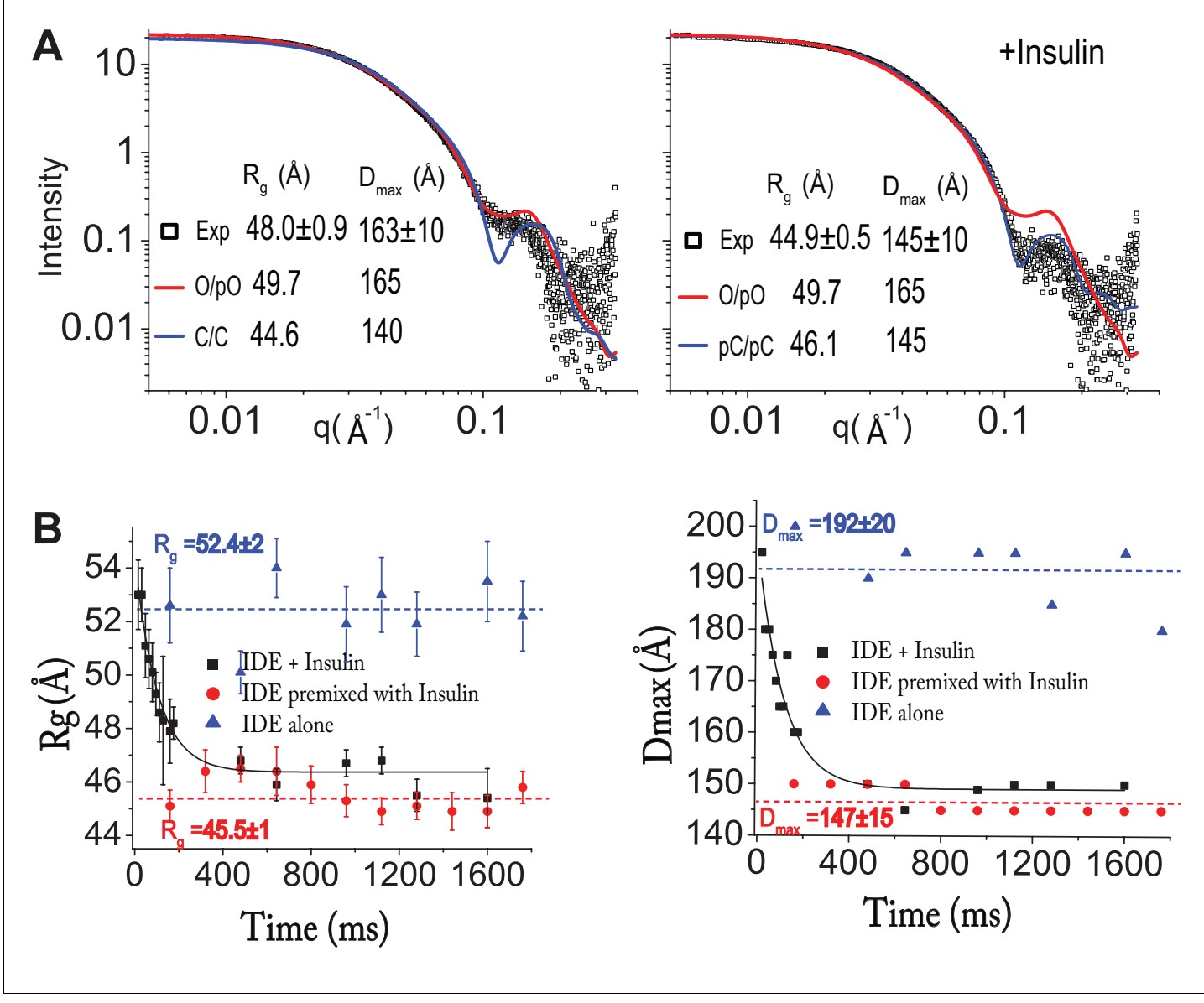

**Figure 4.** Insulin-induced conformational changes of IDE revealed by SAXS. (**A**) Scattering curves from SEC-SAXS analysis using 2–3 mg IDE in the presence (left) or absence (right) of fourfold molar excess of insulin. Solid red and blue lines represent the predicted SAXS curves based on indicated IDE structures. (**B**) Time-resolved SAXS analysis of IDE-insulin interaction. Rapid mixing of IDE with insulin resulted in time-dependent reduction of $R_g$ with $\tau = 0.1$ s.

DOI: https://doi.org/10.7554/eLife.33572.018

The following figure supplement is available for figure 4:

**Figure supplement 1.** SAXS analysis of IDE.
DOI: https://doi.org/10.7554/eLife.33572.019

clearly visible inside the IDE catalytic chamber. This hypothesis also predicts that the IDE door sub-domains in the cryoEM structures of apo-IDE dimer would have the higher thermal B factors than those in that of insulin-bound IDE dimer. Indeed, while clearly visible, the IDE door subdomain, particularly the catalytic zinc-coordinating α4 helix, has significantly higher thermal B factors than in the insulin-bound structure (*Figure 5B*, *Figure 5—figure supplement 2*). The presence of insulin thus profoundly reduces the thermal B factors of IDE door subdomain (*Figure 5B*, *Figure 5—figure supplement 2*). Together, our data support the stabilization of the partial unfolding and/or motion of

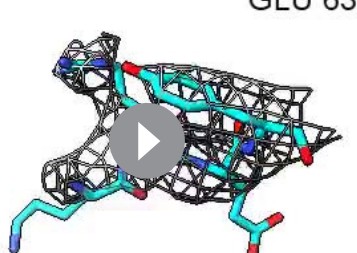

GLU 63

**Video 2.** Fit of individual residues of an IDE subunit of insulin-bound IDE at 3.8 angstrom resolution with the electron density map. The movie is 30 minutes long and is designed for the viewer to zip through residues by the control bar of the movie running software.
DOI: https://doi.org/10.7554/eLife.33572.017

IDE catalytic domain by substrate binding. This lends credence to our hypothesis that IDE catalysis is assisted by its substrates and the importance of this region to the proper functioning of the enzyme.

We then probed the dynamics of IDE door subdomain using two high-affinity IDE inhibitors, BDM44768 and 6bK by HDX-MS. Peptide amide HDX is a powerful tool to probe protein conformational dynamics because it allows evaluation of comparative solvent accessibility throughout the protein (*Chung et al., 2011*; *Deprez-Poulain et al., 2015a*; *Li et al., 2011*; *Maianti et al., 2014*; *Marciano et al., 2014*). These two inhibitors bind different sites to compete with substrate binding. BDM44768 binds the IDE catalytic zinc-binding site while 6bK binds to a site distinct from the catalytic cleft and the N-terminal substrate anchoring exosite. In addition to the expected HDX reduction where BDM44768 and 6bK directly bind, both inhibitors also decreased HDX in the IDE door subdomain (*Figure 5C and D*, *Figure 5—figure supplements 3–6*). These data support the dynamic nature of the catalytic cleft within IDE door subdomain and the importance of this region to the proper functioning of the enzyme.

The combination of HDX-MS with cryoEM structures offers a framework for studying the detailed conformational changes of IDE induced by its substrates. Multiple regions in IDE exhibited a significant reduction in HDX upon insulin binding (*Figure 6A*, *Figure 6—figure supplements 1–4*). These regions correlate well with those that have higher thermal B factors in our cryoEM structures when insulin is absent (*Figure 5B*). These include known substrate-binding sites of IDE: the door subdomain that contains the zinc catalytic site in domain D1, exosite in domain D2 that anchors the N-terminal region of IDE substrates, and residues 821–830 in domain D4 that binds the P1' and P2' residues of IDE substrates after the scissile bond (*McCord et al., 2013*; *Shen et al., 2006*). Insulin also reduced HDX in regions that directly bind IDE door subdomain including the hydrophobic rich H loop, residues 668–673 and residues 821–830 (*McCord et al., 2013*). Together, these data support the conclusion that the binding of substrate stabilizes the IDE catalytic site. Furthermore, insulin binding also reduced the HDX at the IDE-N and IDE-C joining loop and its underlying α-helices, α11 and α20, which is consistent with our SAXS data showing that insulin facilitates the open-closed transition of IDE. A similar pattern of HDX reduction in IDE was also observed when IDE was mixed with Aβ$_{1-40}$ (*Figure 6B*). However, subtle but noticeable differences exist that could aid the search to achieve substrate-selective modulation of IDE. For example, insulin stabilizes the entire IDE door subdomain (residues 170–237) while Aβ only stabilizes the zinc-binding portion of IDE door subdomain (residues 170–203) (*Figure 6—figure supplements 3–4*). Together, our data indicate that the binding of substrates promotes the open to closed transition of IDE and stabilizes the IDE catalytic site for substrate-assisted catalysis.

## Mechanism for the processive degradation of insulin by IDE

IDE processively cuts insulin into two pieces without breaking the disulfide bonds that hold the insulin A and B chains together (*Figure 7A*) (*Manolopoulou et al., 2009*). However, previously reported insulin-bound IDE structures could not explain the processivity of insulin degradation by IDE (*Manolopoulou et al., 2009*). In our cryoEM and crystal structures of insulin-bound IDE, extra electron density was clearly visible inside the IDE catalytic chamber (*Figure 7B and C*, *Figure 7—figure supplement 1*), and the insulin structure is quite different to that reported previously (*Manolopoulou et al., 2009*). The extra density fit well with regions of an unfolded insulin A or B chain with a known scissile bond properly residing at the IDE catalytic site (*Figure 7A and B*, *Figure 7—figure supplement 2*). Additional density is present for part of the other insulin chain, linked by the expected intermolecular disulfide bonds (*Figure 7A*, *Figure 7—figure supplement 2*). We

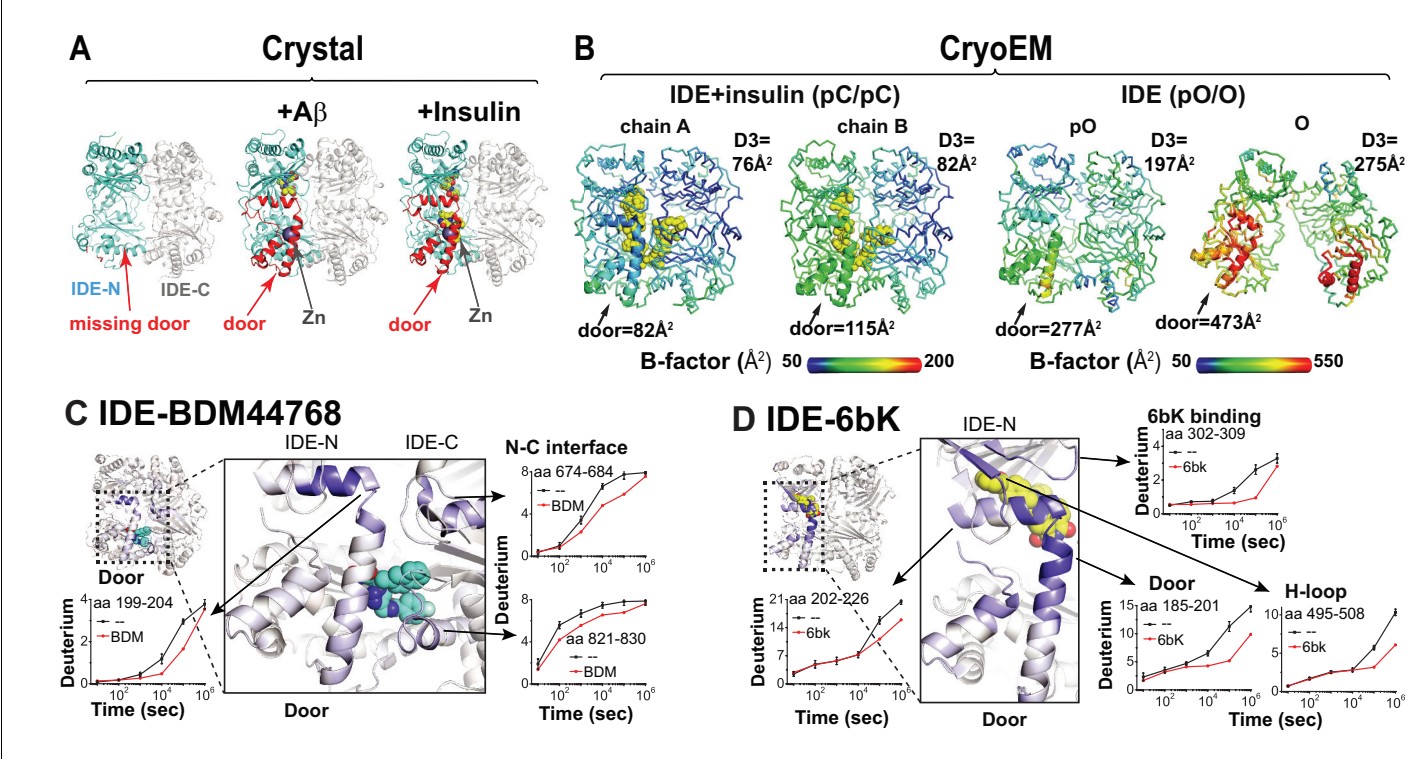

**Figure 5.** Conformational dynamics of IDE door domain. (**A**) Comparison of crystal structures of Fab1-bound IDE alone (PDB code: 4IDF), Fab1- and Aβ-bound IDE (PDB code: 4MIC) and Fab1- and insulin-bound IDE (PDB code: 5WOB). For simplicity, only the subunit of the dimeric IDE that has the profound change in IDE door subdomain is shown and Fab1 is not shown. (**B**) Thermal B factors of two IDE subunits within cryoEM IDE dimer structures of insulin-bound pC/pC state IDE (PDB code: 6B3Q) and pO/O apo-state IDE (PDB code: 6BF8). Averaged B factors of IDE door and D3 are shown for comparison. For simplicity, Fab$_{H11-E}$ is not shown. (**C, D**) Changes in H/D exchange of IDE induced by inhibitors, BDM44768 and 6bK. The changes in H/D are mapped and the progress curve of regions with significant changes are shown.

DOI: https://doi.org/10.7554/eLife.33572.020

The following figure supplements are available for figure 5:

**Figure supplement 1.** Comparison of crystal structures of insulin- or Aβ-bound IDE.
DOI: https://doi.org/10.7554/eLife.33572.021

**Figure supplement 2.** Comparison of thermal B factor distribution of four cryoEM IDE structures, and two crystal structures.
DOI: https://doi.org/10.7554/eLife.33572.022

**Figure supplement 3.** Amide hydrogen-deuterium exchange profiles of IDE alone (**A**) and IDE in the presence of BDM44768 (**B**).
DOI: https://doi.org/10.7554/eLife.33572.023

**Figure supplement 4.** Amide hydrogen-deuterium exchange profiles of IDE alone (**A**) and IDE in the presence of 6bk (**B**).
DOI: https://doi.org/10.7554/eLife.33572.024

**Figure supplement 5.** HDX-MS analysis of IDE-small molecule inhibitor interaction.
DOI: https://doi.org/10.7554/eLife.33572.025

**Figure supplement 6.** Progression curves that have the significant difference in HDX of IDE when 6bK or BDM-44768 (BDM) were present.
DOI: https://doi.org/10.7554/eLife.33572.026

also found that the N-terminus displays density characteristic of phenylalanine 1 of the insulin B chain, while other density corresponds to the ring structure formed by an intramolecular disulfide and a bulky tyrosine 14 side chain at the P1' cleavage site, which are characteristic of the insulin A chain (*Figure 7—figure supplement 2*). Thus, this density has key features that can be contributed by either chain and can be best interpreted as an ensemble of unfolded insulin A and B chains, not belonging solely to either chain.

Together, these structures provide the molecular basis of how IDE could processively cut insulin into two pieces without having to choose which insulin chain to cut first. Upon insulin capture by IDE catalytic chamber, IDE stochastically cuts whichever of either insulin A or B chain binds the IDE catalytic cleft first after insulin unfolding inside the catalytic chamber (*Figure 8A*). Our structures also

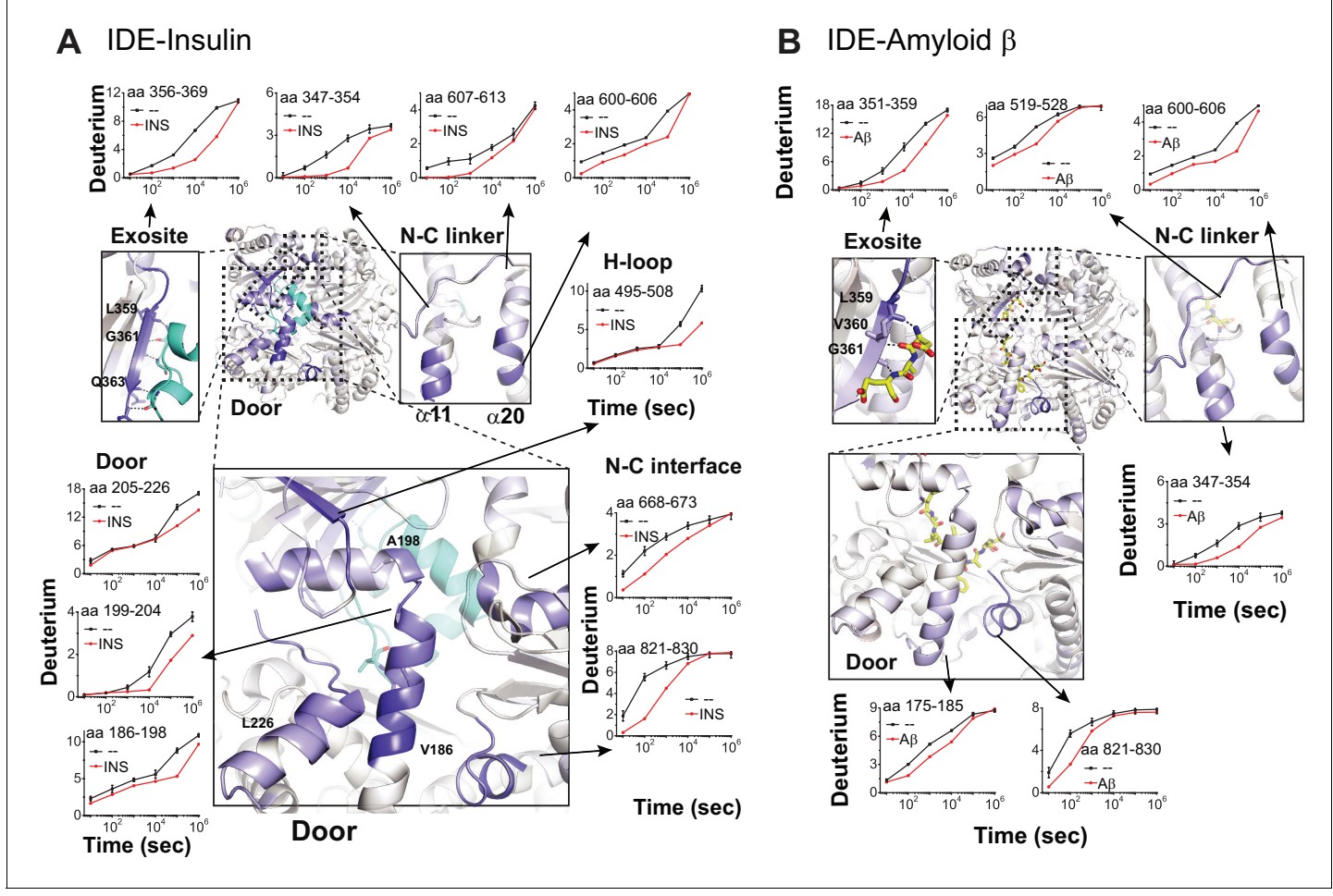

**Figure 6.** Substrate-induced changes in H/D exchange of IDE. (A) Major changes in H/D exchange of IDE induced by insulin. Changes in H/D are mapped onto IDE structure and the progress curve of regions with significant changes are shown. (B) Major changes in H/D exchange of IDE induced by Aβ.

DOI: https://doi.org/10.7554/eLife.33572.027

The following figure supplements are available for figure 6:

**Figure supplement 1.** Amide hydrogen-deuterium exchange profiles of IDE alone (A) and IDE in the presence of insulin (B).
DOI: https://doi.org/10.7554/eLife.33572.028

**Figure supplement 2.** Amide hydrogen-deuterium exchange profiles of IDE alone (A) and IDE in the presence of Aβ (B).
DOI: https://doi.org/10.7554/eLife.33572.029

**Figure supplement 3.** HDX-MS analysis of IDE-substrate interaction.
DOI: https://doi.org/10.7554/eLife.33572.030

**Figure supplement 4.** Progression curves that have the significant difference in HDX of IDE when insulin (INS) or Aβ were present.
DOI: https://doi.org/10.7554/eLife.33572.031

reveal that the N-terminus of insulin interacts extensively with the IDE-N exosite while IDE-C makes substantial contacts with the C-terminal part of insulin. As the N- and C-terminal parts of insulin of the singly-cut insulin are still joined by the inter-molecular disulfide bond, insulin could still effectively keep IDE in the closed conformation. The subsequent unfolding of cleaved insulin leads to processive cleavage of the other insulin chain (*Figure 8A*). The resulting N- and C-terminal insulin fragments would then be released upon the closed to open transition of IDE.

## Discussion

By combining cryoEM, crystallography, SAXS, and HDX-MS, our integrative structural analysis reveals the molecular details of how IDE undergoes open to closed conformational switches for the capture, unfolding, and degradation of insulin and peptides that tend to form amyloid fibrils (*Figure 8B*). By rigid-body motion between IDE-N and IDE-C, IDE switches between O and pO states. This results in three possible conformers where the O/pO dimer is dominant (*Figure 8B*, *Video 3*). Only open-state IDE captures large peptide substrates (e.g. insulin and Aβ). The degree of opening and the charge distribution of the IDE catalytic chamber determine which peptides are captured by IDE depending on their size and high dipole moment (*Figure 3D*). The motions between open- and closed-state IDE, in conjunction with the selective interactions of the IDE catalytic chamber with these peptides, for example, IDE exosite with peptide's N-terminus, creates a force to selectively unfold amyloidogenic peptides. The exposed β-strand of these peptides then stabilizes the inherently unstable IDE catalytic center, leading to the degradation of amyloidogenic peptides by IDE.

The cryoEM structures of open-state IDE suggest the additional ways how IDE may work. Based on the size of catalytic chamber in the closed-state IDE, it is postulated that the entrapment of substrates inside the enclosed catalytic chamber of IDE is required for catalysis (*Malito et al., 2008a*; *Shen et al., 2006*). Thus, IDE only degrades peptides that are capable of fitting into the IDE catalytic chamber. This explains well why all of well-characterized IDE substrates are peptides less than 80 amino acids long (*Malito et al., 2008a*; *Tang, 2016*). This model has successfully guided the identification of CCL3 and CCL4 as novel IDE substrates and the combination of the oligomerization of these chemokines and their degradation by IDE modulates the effectiveness of the chemotactic gradient formed by these chemokines (*Ren et al., 2010*). It does not escape our attention that IDE should be able to degrade larger proteins if such proteins are readily captured by open-state IDE via the charge and surface complementarity and can effectively stabilize IDE catalytic cleft. We also envision that open-state IDE can bind α-synuclein oligomers and alter the kinetics of oligomerization process, which explains how IDE reduces the amyloid fibril formation of α-synuclein in vitro (*Sharma et al., 2015*).

The catalytic activity of IDE is allosterically regulated by its substrate, ATP, and other partner proteins (*McCord et al., 2013*; *Ralat et al., 2011*; *Song et al., 2003*; *Song et al., 2010*; *Tang, 2016*). Based on our data, we put forth a model to explain how the equilibrium between IDE open and 'closed' states facilitates IDE allostery (for simplicity, we group structurally similar pO, pC, and C states into the 'closed' state that is distinct from open-state IDE). By the extensive contacts between IDE-C domains, IDE readily dimerizes (Kd =~ 10 nM) (*Li et al., 2006*; *Shen et al., 2006*). Our cryoEM data reveal that two IDE-N domains within the IDE dimer undergoes rigid body motion, allowing IDE to undergo the transition between the open and 'closed' states without the assistance of substrate (*Figure 8B*). Furthermore, the preferred O/pO combination indicates that the motion of IDE-N in one subunit of IDE dimer is not independent from the other (*Figure 8B*). We thus envision that the preferred combination of the open and 'closed' states within the IDE dimer would allow the substrate-induced closure of one subunit to promote the switch of the other subunit to the open state, allowing cleaved products to be released or substrate captured (*Video 3* synchronized motion 1). Conversely, the opening of one subunit from the 'closed'-state will promote the closure of the other for substrate unfolding and catalysis (*Video 3* synchronized motion 1). Such motions can explain how substrate allosterically regulates IDE activity and how monomerization mutations render IDE less active as well as the lose the ATP- and substrate-mediated regulation (*Ralat et al., 2011*; *Song et al., 2003*; *Song et al., 2010*). This model is also consistent with the kinetic studies which show that phenylalanine 530 mutation to alanine at the linker joining IDE-N and IDE-C makes IDE hyperactive and alters allosteric regulation (*McCord et al., 2013*). Our HDX-MS data reveal that the binding of IDE substrate or inhibitor only affects the deuterium exchange at the IDE-N and IDE interface and the linker between IDE-N and IDE-C, not between the interface of two IDE subunits (*Figures 5* and *6*). Thus, our data does not offer an obvious path for substrate-facilitated allosteric communication between IDE subunits. We speculate that IDE allostery is mediated by the collective motions of many atoms in IDE, not by a subset of atoms within a defined path. The detailed mechanism for IDE allostery awaits future MD simulation studies.

Many, if not most, molecular machines are conformationally heterogeneous, adopting a variety of different structural conformers in solution as they adapt form to serve function. CryoEM is uniquely

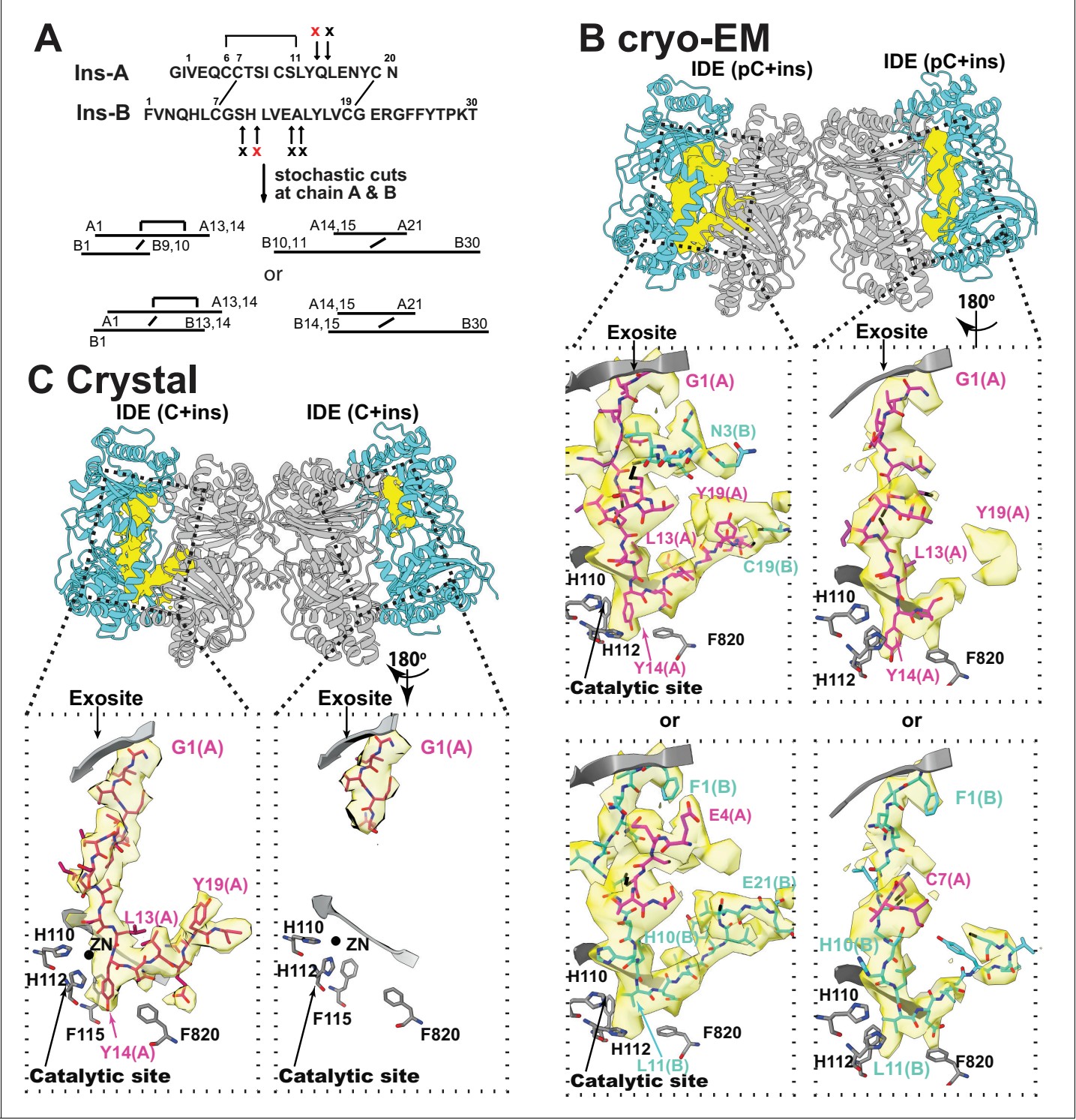

**Figure 7.** The interaction of IDE with insulin. (**A**) Cleavage sites of insulin by IDE. The initial cleavages and resulting fragments are shown. The red X marks the scissile bond revealed by insulin-bound IDE structures. (**B**) CryoEM structure of unfolded insulin inside IDE. The fitting of electron density at the catalytic sites with either insulin A (PDB code: 6B3Q) or B chain (PDB code: 6BFC) are shown. For simplicity, Fab_{H11-E} is not shown. (**C**) Structure of unfolded insulin inside IDE from insulin bound IDE-Fab1 complex (PDB code: 5WOB). For simplicity, only the fitting of insulin A chain is shown.

DOI: https://doi.org/10.7554/eLife.33572.032

The following figure supplements are available for figure 7:

**Figure supplement 1.** Insulin binding at IDE catalytic chamber from Fab1- and insulin-bound IDE crystal structure at 3.95 Å resolution.

*Figure 7 continued on next page*

*Figure 7 continued*

DOI: https://doi.org/10.7554/eLife.33572.033

**Figure supplement 2.** The fit of the unfolded insulin A or B chain with electron density inside the catalytic chamber of IDE from cryoEM structure of insulin-bound IDE.

DOI: https://doi.org/10.7554/eLife.33572.034

capable of solving the structures of these large flexible macro-molecules. Improvements in the hardware technology over the past 5 years (*Li et al., 2013*) now provide for excellent quality images and improvements in software allow for classification of particles of different conformations from a heterogeneous mixture of structures (*Scheres, 2012*). Our approach to obtain cryoEM structure of the open-state IDE, a key functional state recalcitrant to crystallization, is generally applicable to other proteins. In addition to its relatively small size (220 kDa), the IDE dimer adopts multiple conformational states to fulfill its function. To solve the challenge posed by size and pseudo two-fold symmetry in our system, we included high-affinity IDE-binding Fabs, which increases the size of the complex, breaks the apparent symmetry, and provides excellent fiducials for validating the morphology and resolution of the overall structure. We take advantage of two other technological developments to improve cryoEM structure determination of the IDE dimer. A new vitrification device, Spotiton, allows us to prepare samples embedded in a very thin and even layer of vitrified ice, which improves data quality (*Dandey et al., 2018*; *Razinkov et al., 2016*). We also addressed the issues of preferred orientation, quite common for many samples, by acquiring images from gold-coated grids tilted relative to the electron beam (*Russo and Passmore, 2014*; *Tan et al., 2017*). In conjunction with the state-of-art hardware and software in data acquisition, these implementations allowed us to obtain near atomic resolution 3D structures of the open-state IDE despite its conformational heterogeneity.

The accumulation of toxic amyloid fibrils is associated with many human diseases and IDE plays a key role in preventing amyloid fibril formation by its proteolytic activities (*Jucker and Walker, 2013*; *Tang, 2016*). Accumulating data also suggest that IDE plays non-proteolytic roles to regulate other proteostatic processes, for example, regulating proteasome activity and preventing amyloid fibril formation of α-synuclein (*Tang, 2016*). As IDE modulates proteostasis by targeting diverse proteins, substrate selective modulation of IDE activity is crucial to realize IDE-based therapy (*Pivovarova et al., 2016*; *Tang, 2016*). Indeed, noticeable differences in the reduction of HDX between insulin- and Aβ-bound IDE exist (*Figure 6A and B*). Together with our cryoEM structures, our studies offer a road map to develop insulin-selective inhibitors or Aβ-selective enhancers to treat diabetes and Alzheimer's disease.

# Materials and methods

**Key resources table**

| Reagent type (species) or resource | Designation | Source or reference | Identifiers | Additional information |
|---|---|---|---|---|
| Antibody | Synthetic anti-IDE antibody Fab fragment (Fab$_{H11-E}$) | This study | NA | About 5 mg/ml for crystallization, and 0.15 mg/ml for cryoEM. |
| Antibody | Synthetic anti-IDE antibody Fab fragment Fab1Fab1 | (*McCord et al., 2013*) | NA | About 5 mg/ml for crystallization. |
| Recombinant protein (Human) | Cysteine-free IDE | This study | NA | Described as above. |
| Recombinant protein (Human) | Cysteine-free IDE-E111Q | This study | NA | Described as above. |
| Recombinant protein (Human) | Insulin | Sigma-Aldrich | 234-279-7 | |
| Recombinant protein (Human) | Amyloid β | (*King et al., 2014*) | NA | |

*Continued on next page*

*Continued*

| Reagent type (species) or resource | Designation | Source or reference | Identifiers | Additional information |
|---|---|---|---|---|
| Recombinant protein (Human) | Pepsin | Sigma-Aldrich | P6887-1G | |
| Chemical compound, drug | BDM44768 | (*Deprez-Poulain et al., 2015a*) | NA | |
| Chemical compound, drug | 6bK | (*Maianti et al., 2014*) | NA | |
| Chemical compound | D2O | Cambridge Isotope Laboratories, Inc. | DLM-4–1L | |
| Software, algorithm | Leginon 3.3 | (*Suloway et al., 2005*) | NA | http://emg.nysbc.org/redmine/projects/leginon/wiki/Leginon_Homepage |
| Software, algorithm | DoGpicker | (*Voss et al., 2009*) | NA | http://emg.nysbc.org/redmine/projects/software/wiki/DoGpicker |
| Software, algorithm | MotionCor2 | (*Zheng et al., 2017*) | NA | http://msg.ucsf.edu/em/software/motioncor2.html |
| Software, algorithm | Relion2.0 | (*Scheres, 2012*) | NA | http://www2.mrc-lmb.cam.ac.uk/relion/index.php/Main_Page |
| Software, algorithm | Relion2.1 | (*Scheres, 2012*) | NA | http://www2.mrc-lmb.cam.ac.uk/relion/index.php/Main_Page |
| Software, algorithm | 3DFSC | (*Tan et al., 2017*) | NA | https://github.com/nysbc/Anisotropy |
| Software, algorithm | UCSF ChimeraX | (*Goddard et al., 2018*) | NA | https://www.cgl.ucsf.edu/chimerax/ |
| Software, algorithm | UCSF Chimera | (*Pettersen et al., 2004*) | NA | https://www.cgl.ucsf.edu/chimera |
| Software, algorithm | PHENIX | (*Adams et al., 2010*) | NA | https://www.phenix-online.org/ |
| Software, algorithm | Phaser-MR | (*McCoy et al., 2007*) | NA | https://www.phenix-online.org/documentation/tutorials/mr.html |
| Software, algorithm | COOT | (*Emsley et al., 2010a*) | NA | https://www2.mrc-lmb.ac.uk/personal/pemsley/coot |
| Software, algorithm | MolProbity | (*Chen et al., 2010*) | NA | http://www.ks.uiuc.edu/Research/mdff/ |
| Software, algorithm | PRIMUS | (*Konarev et al., 2003*) | NA | https://www.embl-hamburg.de/biosaxs/primus.html |
| Software, algorithm | CRYSOL | (*Svergun et al., 1995*) | NA | https://www.embl-hamburg.de/biosaxs/crysol.html |
| Software, algorithm | GNOM | (*Svergun, 1992*) | NA | https://www.embl-hamburg.de/biosaxs/gnom.html |
| Software, algorithm | SEQUEST/Proteome Discoverer Software | ThermoFisher Scientific | NA | https://www.thermofisher.com/order/catalog/product/OPTON-30795 |
| Software, algorithm | HDEXaminer | Sierra Analytics | NA | http://massspec.com/hdexaminer/ |

## Protein

Wild-type human IDE, cysteine-free IDE (IDE-CF), and catalytically inactive IDE mutant, IDE-CF-E111Q were expressed in *E. coli* BL21 (DE3) cells (at 25°C and 20 hr, 0.5 mM IPTG induction using T7 medium). His-tagged, biotinylated IDE was expressed in *E. coli* BL21 (DE3) that carried two plasmids, one for IDE with an N-terminal His-tag and a C-terminal AviTag (GLNDIFEAQKIEWHE), and the other for *E. coli* BirA, a biotin ligase that transfers biotin to AviTag. Recombinant IDE proteins were purified by Ni-NTA, source-Q, and Superdex 200 columns as previously described (*Manolopoulou et al., 2009*). Aβ$_{1-40}$ was synthesized at a 0.25 mmol scale using Fmoc and HBTU/HOBt chemistry on an Applied Biosystems 433A instrument and purified by RP-HPLC, lyophilized, and stored at −20°C under Argon as described (*King et al., 2014*; *Sohma et al., 2004*). Insulin was purchased from SIGMA (91077C).

## IDE-binding synthetic antibody

We screened antigen-binding fragments (Fab) from a phage-display library using immobilized biotinylated IDE as previously described (*Miller et al., 2012*). To obtain Fabs binding to an epitope distinct from the previously reported IDE-Fab named Fab1 (*McCord et al., 2013*), we used 1 μM Fab1 as a competitor for panning in the final round of selection. In total, 18 new IDE binding Fabs were isolated and characterized. Of those, IDE-Fab$_{H11}$ was used in structural studies. Elbow-enhanced Fab1 (Fab1$_E$) and Fab$_{H11-E}$ were engineered to modify the switch residue region of the Fab heavy chain (wild-type sequence: $^{109}$VTVSSASTKGP$^{119}$) to VTVFNQIKGP (*Bailey et al., 2018*). Fabs were expressed in *E. coli* strain BL21(DE3) or 55244 and purified using a HiTrap protein-G HP column or Protein-G-A1 column as described (*Bailey et al., 2014*; *McCord et al., 2013*). Surface plasmon resonance measurements were carried out at 20°C on a Biacore 3000 by immobilizing His-tagged IDE onto a Ni-NTA chip (GE Healthcare) and then injecting 3.3–100 nM of the Fab at a flow rate of 30 μl/min as previously described (*Koide et al., 2012*; *Zhang et al., 2012*).

## Enzymatic activity assays

A fluorogenic bradykinin-mimetic substrate of IDE, substrate V (7-methoxycoumarin-4-yl-acetyl-RPPGF-SAFK-2,4-dinitrophenyl, R and D Systems), was used to measure the enzymatic activity of IDE on a Tecan Safire microplate reader using an excitation wavelength of 327 nm and emission wavelength of 395 nm (*McCord et al., 2013*). Reactions were carried out at 37°C, using 5 μM substrate V in 100 μl of 50 mM potassium phosphate (pH 7.3) with the addition of 0.6 nM IDE in the presence or absence of 12 nM Fab$_{H11}$. The degradation of substrate V was assessed by monitoring fluorescence increase and the initial velocity was calculated using linear regression after background subtraction. The standard deviation was derived from three individual experiments.

## Crystallography

IDE-CF-E111Q was first incubated with Aβ or insulin in a 1:10 or 1:2 ratio, respectively, at 18°C overnight (~16–18 hr) and IDE-substrate complex was purified using Superdex 200 column. This process was repeated three times. Substrate-bound IDE was then mixed in an equimolar ratio with Fab1 or Fab1$_E$ and substrate- and Fab-bound IDE was purified using Superdex 200 column. The resulting complexes were crystallized in 0.1M sodium cacodylate (pH6.5), 0.2M MgCl$_2$, 10% PEG-3000, and 0.01% ethyl acetate at 18°C by hanging drop vapor diffusion. Crystals formed within 2–3 days. To crystallize Fab$_{(H11-E)}$-bound IDE, IDE-CF was purified by Superdex 200 three times before mixing with extra molar Fab$_{H11-E}$ and Fab$_{H11-E}$-bound IDE was purified by Superdex 200. Such protein complex was crystallized in 0.088M Ammonium citrate tribasic, pH 7, 10% w/v PEG3350, 0.02M ethylenediaminetetraacetic disodium salt dihydrate at 18°C by hanging drop vapor diffusion. Crystals formed in about a week. For data collection, crystals were equilibrated in reservoir buffer with 30% glycerol and flash frozen in liquid nitrogen. Diffraction data were collected at 100K on the 19-ID beamline at Argonne National Laboratory. Data sets were processed using HKL2000 and the CCP4 suite. The structures of were determined by molecular replacement. For substrate-bound Fab1-bound IDE structure, the unbound IDE-Fab$_1$ complex (4IOF) was used as a search model and no NCS average for C2 symmetry of IDE dimer was applied during the refinement to avoid the bias. For the crystal structure of Fab$_{H11-E}$-bound IDE, the closed-state IDE (2WBY) and Fab in unbound IDE-Fab$_1$ complex (4IOF) were used as the search model. Model building and refinement were performed by using REFMAC, PHENIX, and COOT (*Adams et al., 2011*; *Emsley et al., 2010b*; *Potterton et al., 2002*). The final model for Aβ-IDE-Fab1 (pdb = 4M1C) has R$_{work}$ = 23% and R$_{free}$ = 27%, that for Insulin-IDE-Fab1 (pdb = 5 WOB) has R$_{work}$ = 24% and R$_{free}$ = 29% and that for IDE-Fab$_{H11-E}$ (pdb = 5 UOC) has R$_{work}$ = 22% and R$_{free}$ = 27%. The data and refinement statistics are listed on *Supplementary file 1*. The key residues involving in the interaction of IDE with Fab$_{H11-E}$ was calculated using PDBePISA (*Krissinel and Henrick, 2007*).

## CryoEM

Purified wild-type IDE was further purified by Superdex 200 chromatography using buffer containing 20 mM HEPES, pH 7.2, 300 mM NaCl, and 20 mM EDTA and then mixed with Fab$_{H11-E}$ at an equal molar ratio. Fab$_{H11-E}$-IDE complex was purified by Superdex 200 chromatography in the absence or presence of five-fold molar excess of insulin. Insulin-bound IDE-Fab$_{H11-E}$ and IDE-Fab$_{H11-E}$ cryoEM

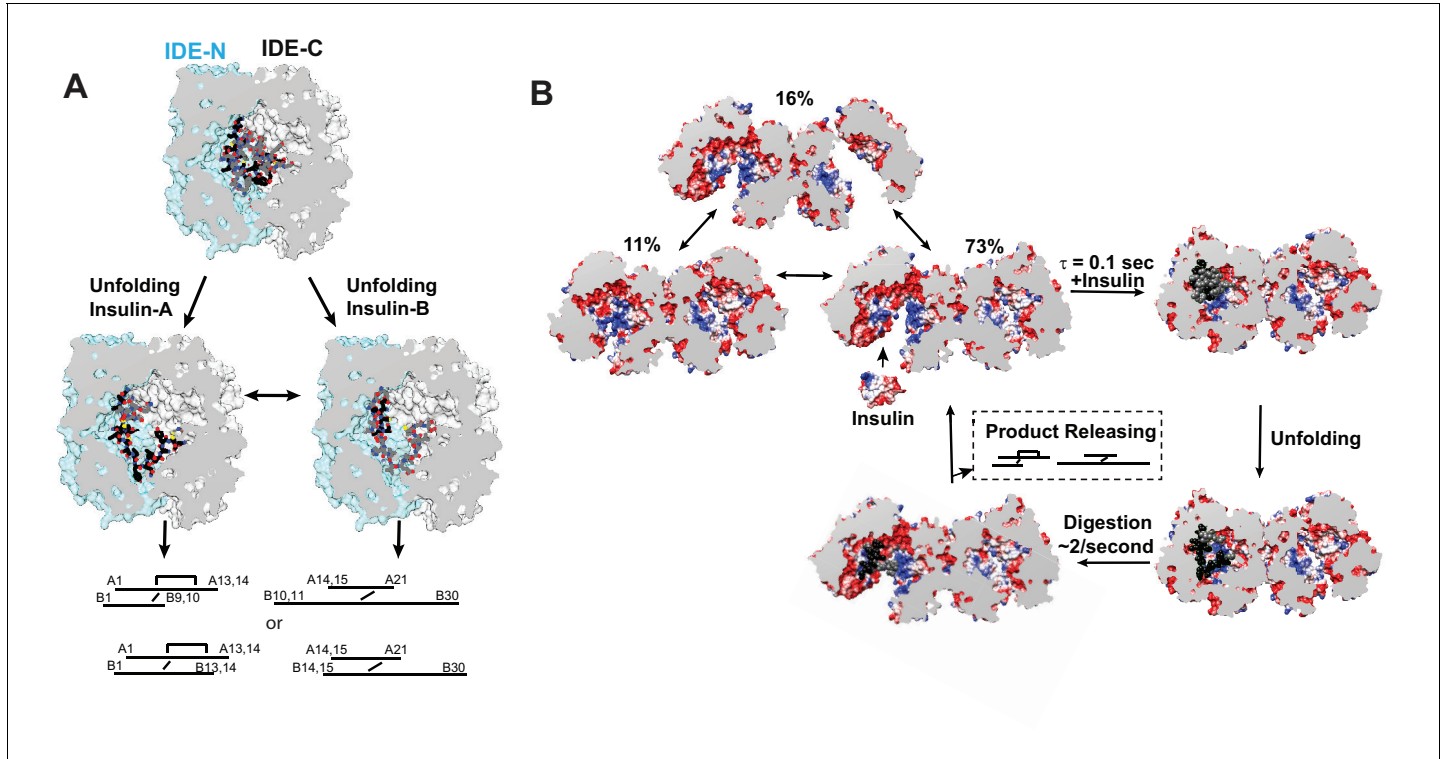

**Figure 8.** Models for IDE catalytic cycle. (**A**) Model for conformational changes of insulin inside IDE chamber. Structure of partially unfolded insulin at the IDE catalytic chamber is from PDB code 2WBY while those with unfolded insulin are from PDB code 6B3Q and 6BFC for insulin A and B chain at IDE catalytic site. (**B**) A model for IDE conformational states relevant to substrate capture and catalysis. For simplicity, only the capture and catalysis of the dominant IDE conformational state, pO/O, is shown. The conformational change of the other IDE subunit allosterically regulated by insulin binding to the open-state IDE within an IDE dimer depicted by *Video 3* is not shown. The electrostatic surface potential of IDE is set from −5 kT/e (red) to 5 kT/e (blue) and calculated by UCSF Chimera, and the section interface is colored as grey.

DOI: https://doi.org/10.7554/eLife.33572.035

grids were prepared using either a manual plunger or Spotiton 1.0 (*Dandey et al., 2018*; *Razinkov et al., 2016*). For manual plunging, 300 mesh carbon or gold lacey grids, prepared in house (*Fukami and Adachi, 1965*), were plasma cleaned using $O_2$ and $H_2$ for 30 s using a Solarus plasma cleaner (Gatan). 3 µl of sample was applied to the grid and manually blotted with filter paper for 3 s from the back of the grids followed immediately by plunging into liquid ethane. For Spotiton prepared grids, 300 mesh carbon or gold lacey nanowire grids were plasma cleaned with $O_2$ and $H_2$ for 10 secs using a Solarus plasma cleaner (Gatan). 6 µl of protein sample was aspirated by the Spotiton piezo tip and about $80 \times 50$ pl of sample was delivered to the grid as it passes the piezo tip en route to being plunged into liquid ethane. All images were acquired using a Titan Krios microscope (FEI) operated at 300KeV with a Gatan K2 direct electron detector (Gatan) in counting mode. Images were automatically acquired using Leginon (*Suloway et al., 2005*) using collection parameters as shown in *Supplementary file 2*. IDE-Fab$_{H11-E}$ showed highly preferred orientation in vitreous ice. Thus, images were acquired with the grid tilted at an angle to the electron beam, which allowed better sampling of other orientations (*Figure 2—figure supplement 9B*) (*Tan et al., 2017*). Images were processed using software integrated into the Appion (*Lander et al., 2009*) pipeline except where stated. Frames were aligned using MotionCor2 software with dose weighting (*Zheng et al., 2017*), particles were picked and extracted automatically using DoGpicker (*Voss et al., 2009*). Particle stacks were then passed to RELION2.1 (*Scheres, 2012*) and processed through several rounds of 2D and 3D classification. Example images and 2D class averages are shown in *Figure 2—figure supplements 5–7*. Selected classes were then processed for high resolution 3D refinement (*Figure 2—figure supplement 8*). The resolution was further improved by focused classification with signal subtraction. The mask was created in Chimera (*Pettersen et al., 2004*) with the Fab regions set to value

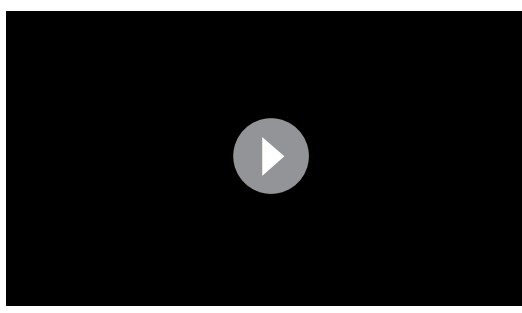

**Video 3.** Morph between IDE conformational states to depict the possible transition between various conformational states of IDE. Only a few synchronized motions are depicted in such morph. Uncoordinated motions between two IDE subunits are not depicted even though they likely occur as well. Scheme 1: Morph to depict synchronized motion of IDE that one IDE subunit undergoes the transition from the open state to the partially open state while the other undergoes the transition in the opposite direction (from the partially open state to the open state). Scheme 2: Morph to depict synchronized motion of IDE that both IDE subunits undergoes the transition between the open state and the partially open state simultaneously. Scheme 3:Morph to depict synchronized motion of IDE that both IDE subunits undergoes the transition between the partially-open state and the partially closed state simultaneously. Scheme 4: Morph to depict synchronized motion of IDE that IDE subunits simultaneously undergoes the transition between the partially-open state, the closed state, and the partially closed state. We hypothesize that IDE undergoes such motion when its substrates are entrapped inside the catalytic chamber. Such motion would allow substrate unfolding, leading the substrate-assisted catalysis.
DOI: https://doi.org/10.7554/eLife.33572.036

zero and outside Fab regions set to value one. Subsequently the masked refinement was carried out with C1 symmetry using RELION 2.1 (*Scheres, 2012*). The final resolution was estimated using Fourier Shell Correlation (FSC = 0.143) in RELION 2.1 (*Figure 2—figure supplement 9C and D*). The anisotropy was analyzed using 3DFSC (*Tan et al., 2017*). The cryoEM data collection and processing statistics are listed in *Supplementary file 2*. The structural models were built using a combination of high-resolution IDE dimer structure (PDB = 2 cww) and $Fab_{H11-E}$-bound IDE dimer. The density fitting and structure refinement was done using UCSF CHIMERA (*Pettersen et al., 2004*), COOT (*Emsley and Cowtan, 2004*; *Emsley et al., 2010b*), and PHENIX (*Adams et al., 2011*). The refinement statistics are listed in *Supplementary file 3*.

## SAXS

SAXS data were collected at the BioCAT/18ID beamline at Advanced Photon Source, Argonne National Laboratory (Chicago, USA) (*Fischetti et al., 2004*) using the photon counting PILATUS 3 1M at room temperature ($23^{\circ}$C) and an incident X-ray wavelength of 1.03 Å. The 3.5 m sample-to-detector distance yielded a range of 0.005–0.33 $Å^{-1}$ for the momentum transfer ($q = 4\pi \sin\theta/\lambda$ where $2\theta$ is the scattered angle between the incident and scattered beam and $\lambda$ the X-ray wavelength). For size exclusion chromatography (SEC)-SAXS, 2–3 mg IDE was loaded onto a GE Superdex 200 10/300G in the buffer containing 20 mM Tris, pH 8.0, 100 mM NaCl and the eluates were immediately passed through X-ray beam. To prepare zinc-free catalytic inactive IDE, IDE was first washed with about

2M NaCl and then dialyzed against 500 ml 20 mM Tris pH8.0, 100 mM NaCl, 20 mM EDTA and subsequently purified by Superdex 200 column using the buffer containing 20 mM Tris, pH 8.0, 100 mM NaCl, 20 mM EDTA. Greater than 15 measurements around IDE protein peak were collected in each run for subsequent analysis. Time-resolved SAXS experiments were done using a microfluidic laminar flow mixer adapted from the design initially made for time resolved fluorescence studies (*Park et al., 2008*; *Park et al., 2006*). The mixer chip consisted of 5 inlets: a center channel for IDE solution, two diagonal channels for identical buffers as the IDE solution and two side channels for buffer containing insulin. Syringe pumps (Model 511, New Era Pump Systems Inc., NY) were used to inject sample and buffer solutions continuously into the mixer inlets. Per measurement, 5–10 mg IDE injected from the center channel were compressed into a thin sheet by the diagonal and side channel flows as a result of the hydrodynamic focusing effect. Kinetic reaction was triggered by diffusion of twofold molar excess of insulin into the jet of IDE from adjacent flow layers of buffer above and below. The thickness of the sheet was controlled by the flow rate ratios of the inlet channels. The observation channel is 1 mm deep (i.e. X-ray pathlength), 0.2 mm wide and 25 mm long. During the flow experiment, the observation channel was sampled by a grid scan of 10 × 30 positions and SAXS data was collected at each grid point. The total flow rate determined the maximum accessible time corresponding to the exit end of the observation channel. The time window of 2 s was performed by using the flow rates of 68, 2, 9 µL/min for side, diagonal and center syringe

pumps, respectively. Increasing the flow rates to 682, 22 and 90 µL/min allows the kinetic study in the time window shorter than 0.2 s. We did three mixing experiments: i) IDE bound with insulin mixing with insulin buffer; ii) IDE mixing with insulin free buffer; and iii) IDE mixing with buffer containing insulin. The mixing experiments i and ii were done using flow rates for 2 s time window while the experiment iii was done for both 2 s and 0.2 s time windows. The SAXS data reduction and analyses were done using ATSAS package (*Petoukhov et al., 2012*), PRIMUS (*Konarev et al., 2003*) and CRYSOL (*Svergun et al., 1995*). Comparison of experimental scattering profiles with calculated profiles from high-resolution atomic models was done using CRYSOL (*Svergun et al., 1995*). For time-resolved experiments, in order to reduce the effects of the varied parasitic scattering in grid scans, radius of gyration values were obtained using the program GNOM (*Petoukhov et al., 2012*).

## Hydrogen-deuterium exchange mass spectrometry

Prior to carrying out hydrogen/deuterium exchange experiments, the optimal quench condition that generated the best sequence coverage map of IDE were established as previously described (*Marsh et al., 2013*). IDE stock solutions were prepared in 8.3 mM Tris-HCl, 50 m NaCl, 2.1% DMSO, pH7.2 $H_2O$ buffer by mixing 15 µM of IDE with 30 µM $Fab_{H11-E}$, 150 µM of Insulin, 150 µM of 6bk, 150 µM of BDM44768 or 300 µM of Aβ, incubating at room temperature for 30 min, then keep on ice for 15 min. Functional hydrogen-deuterium exchange reactions were initiated by dilution of 3 µl of stock solution into 9 µl of $D_2O$ buffer (8.3 mM Tris, 50 mM NaCl, pDREAD 7.2) and incubation at 0°C. The exchange reactions were quenched after various exchange time points (10, 100, 1000, 10,000, 100,000 s at 0°C, and 100,000 s at RT) by adding 18 µl of ice-cold 0.8% formic acid, 1.6M GuHCl, 16.6% glycerol for a final pH of 2.5. Quenched samples were then immediately frozen on dry ice and stored at −80°C before LC/MS analysis. Non-deuterated and equilibrium-deuterated control samples are also prepared as previously described (*Tsalkova et al., 2012*). The frozen samples were thawed automatically on a cryogenic autosampler(*Woods and Hamuro, 2001*) at 4°C, and digested on an immobilized pepsin column (16 µl bed volume). Proteolytic products were collected on a C18 trap column (Michrom Magic C18 AQ 0.2 × 1 mm) for 1 min desalting and separated using a reverse phase analytical column (Michrom Magic C18 AQ 0.2 × 50 mm, 3 µm) with a acetonitrile linear gradient (6.4%–38.4% over 30 min). MS analysis was performed on an OrbiTrap Elite Mass Spectrometer (ThermoFisher Scientific, San Jose, CA). Instruments settings was optimized to minimize the back-exchange(*Walters et al., 2012*). The data was acquired in both data-dependent MS/MS mode and MS1 profile mode and Proteome Discoverer software (ThermoFisher) was used to identify proteolytic peptides. The deuterium content of the peptides for each time point was calculated by DXMS Explorer (Sierra Analytics Inc, Modesto, CA), with corrections for back-exchange (*Zhang and Smith, 1993*). H/D exchange experiments performed using our automated system typically produce deuteron incorporation measurements with a standard deviation of less than 2% of the mean of triplicate determinations (*Chang et al., 2015*; *Hsu et al., 2009a*; *Hsu et al., 2009b*). In the present work, just as in our previous studies, only changes in deuteration level greater than 10% were considered significant (*Burke et al., 2009*; *Hamuro et al., 2004*). Heat maps were created using in house program that sub-localize HDX down to smaller fragments using all overlapping peptides.

## Data and software availability

The cryoEM maps have been deposited in the Electron Microscopy Data Bank with accession codes EMD-7041, EMD-7092, EMD-7065, EMD-7090, EMD-7062, EMD-7093, EMD-7066, and EMD-7091. The atomic models have been deposited in the Protein Data Bank under accession code EMDB 6B3Q, 6BFC, 6BF8, 6B7Y, 6BF6, 6B70, 6BF9, 6B7Z and 6BF7. The crystal structures of Fab1-bound IDE in complex with insulin or Aβ have been deposited in the Protein Data Bank under access code 5WOB and 4M1C, respectively. The crystal structure of $Fab_1$-bound IDE in complex with insulin and $Fab_{1E}$ has been deposited in the Protein Data Bank under access code 5CJO. The crystal structures of IDE in complex with $Fab_{H11-E}$ has been deposited in the Protein Data Bank under access code 5UOE.

## Acknowledgements

We are grateful to Ryan M Hoffman in Andrew Ward's lab at the Scripps Research Institute for initial negative stained EM data acquisition and analysis, to Srinivas Chakravarthy at BioCAT, APS for assisting with SAXS data collection and analysis, to Juan P Maianti for synthesizing 6bK, and to Steve Meredith for synthetic Aβ. This work was supported by the NIH grants GM81539 and GM121964 to Wei-Jen Tang, GM103310 to Bridget Carragher and Clinton S Potter, Agency for Science, Technology and Research Singapore to Yong Zi Tan, GM103622 to Tom Irving at BioCAT, APS, and DARPA Fold Fx program (N66001-14-2-4053), NIH R35 GM118062, and the Howard Hughes Medical Institute to David Liu. The Simons Electron Microscopy Center is supported by a grant from the Simons Foundation (349247). Use of the Advanced Photon Source was supported by the U.S. Department of Energy, Office of Basic Energy Sciences, under contract No. DE-AC02-06CH11357.

## Additional information

### Funding

| Funder | Grant reference number | Author |
| --- | --- | --- |
| Agency for Science, Technology and Research | | Yong Zi Tan |
| Defense Advanced Research Projects Agency | N66001-14-2-4053 | David R Liu |
| National Institutes of Health | R35 GM118062 | David R Liu |
| Howard Hughes Medical Institute | | David R Liu |
| Simons Foundation | 349247 | Bridget Carragher Clinton S Potter |
| National Institutes of Health | GM103310 | Bridget Carragher Clinton S Potter |
| National Institutes of Health | GM81539 | Wei-Jen Tang |
| National Institutes of Health | GM121964 | Wei-Jen Tang |

The funders had no role in study design, data collection and interpretation, or the decision to submit the work for publication.

### Author contributions

Zhening Zhang, Conceptualization, Data curation, Formal analysis, Investigation, Methodology, Writing—original draft, Writing—review and editing, Performed EM grid preparation, data acquisition and processing; Wenguang G Liang, Conceptualization, Data curation, Formal analysis, Investigation, Methodology, Writing—original draft, Writing—review and editing, Purified protein for EM and conducted negative stained EM data acquisition, Built and refined cryoEM structural models, Performed protein purification and crystallographic data collection, Built and refined structural models, Performed protein purification for HDX-MS, Purified proteins for SAXS studies, Performed SAXS studies; Lucas J Bailey, Data curation, Formal analysis, Investigation, Methodology, Writing—original draft, Writing—review and editing, Screened, purified, and engineered Fab; Yong Zi Tan, Data curation, Formal analysis, Investigation, Methodology, Writing—review and editing, Assisted with EM grid preparation, data acquisition and processing; Hui Wei, Formal analysis, Investigation, Methodology, Assisted with EM grid preparation, data acquisition and processing; Andrew Wang, Data curation, Formal analysis, Writing—review and editing, Built and refined structural models, Performed HDX-MS and analysis; Mara Farcasanu, Data curation, Formal analysis, Investigation, Writing—review and editing, Built and refined structural models; Virgil A Woods, Data curation, Formal analysis; Lauren A McCord, Data curation, Formal analysis, Investigation, Performed protein purification and crystallographic data collection, Built and refined structural models; David Lee, Data curation, Formal analysis, Performed HDX-MS and analysis, Provided critical reagents; Weifeng Shang, Data curation, Formal analysis, Performed SAXS studies; Rebecca Deprez-Poulain, Benoit Deprez, Akiko Koide,

Shohei Koide, Anthony A Kossiakoff, Resources, Provided critical reagents; David R Liu, Resources, Performed HDX-MS and analysis, Provided critical reagents; Sheng Li, Resources, Data curation, Formal analysis, Supervision, Validation, Investigation, Visualization, Methodology, Writing—original draft, Writing—review and editing, Designed the project, Performed HDX-MS and analysis; Bridget Carragher, Clinton S Potter, Conceptualization, Resources, Formal analysis, Supervision, Funding acquisition, Validation, Investigation, Visualization, Writing—original draft, Writing—review and editing, Designed the project, Oversaw EM grid preparation, data acquisition and processing; Wei-Jen Tang, Conceptualization, Resources, Data curation, Formal analysis, Supervision, Funding acquisition, Validation, Investigation, Visualization, Methodology, Writing—original draft, Project administration, Writing—review and editing, Designed the project, Built and refined cryoEM structural models, Built and refined structural models

## Author ORCIDs

Yong Zi Tan (iD) http://orcid.org/0000-0001-6656-6320
Bridget Carragher (iD) https://orcid.org/0000-0002-0624-5020
Clinton S Potter (iD) https://orcid.org/0000-0003-2394-0831
Wei-Jen Tang (iD) https://orcid.org/0000-0002-8267-8995

## Decision letter and Author response

Decision letter https://doi.org/10.7554/eLife.33572.085
Author response https://doi.org/10.7554/eLife.33572.086

# Additional files

## Supplementary files

• Supplementary file 1. Data collection and structure refinement statistics
DOI: https://doi.org/10.7554/eLife.33572.037

• Supplementary file 2. CryoEM data collection and processing statistic
DOI: https://doi.org/10.7554/eLife.33572.038

• Supplementary file 3. CryoEM map and model refinement statistics
DOI: https://doi.org/10.7554/eLife.33572.039

• Supplementary file 4. Distances and angles between center of mass of different domains of IDE cryoEM and crystal structures.
DOI: https://doi.org/10.7554/eLife.33572.040

• Transparent reporting form
DOI: https://doi.org/10.7554/eLife.33572.041

## Major datasets

The following datasets were generated:

| Author(s) | Year | Dataset title | Dataset URL | Database, license, and accessibility information |
|---|---|---|---|---|
| Tang WJ | 2018 | IDE with insulin map | http://www.ebi.ac.uk/pdbe/entry/emdb/EMD-7041 | Publicly available at the Electron Microscopy Data Bank (accession no. EMD-7041) |
| Tang WJ | 2018 | IDE Open(O)/partial-Open(pO) state map | http://www.ebi.ac.uk/pdbe/entry/emdb/EMD-7092 | Publicly available at the Electron Microscopy Data Bank (accession no. EMD-7092) |
| Tang WJ | 2018 | IDE O/O state map | http://www.ebi.ac.uk/pdbe/entry/emdb/EMD-7065 | Publicly available at the Electron Microscopy Data Bank (accession no. EMD-7065) |

| Tang WJ | 2018 | IDE pO/pO state map | http://www.ebi.ac.uk/pdbe/entry/emdb/EMD-7090 | Publicly available at the Electron Microscopy Data Bank (accession no. EMD-7090) |
|---------|------|---------------------|-----------------------------------------------|------------------------------------------------------------------------------------|
| Tang WJ | 2018 | IDE with insulin and FabH11-E map | http://www.ebi.ac.uk/pdbe/entry/emdb/EMD-7062 | Publicly available at the Electron Microscopy Data Bank (accession no. EMD-7062) |
| Tang WJ | 2018 | IDE O/pO state with FabH11-E map | http://www.ebi.ac.uk/pdbe/entry/emdb/EMD-7093 | Publicly available at the Electron Microscopy Data Bank (accession no. EMD-7093) |
| Tang WJ | 2018 | IDE O/O state with FabH11-E map | http://www.ebi.ac.uk/pdbe/entry/emdb/EMD-7066 | Publicly available at the Electron Microscopy Data Bank (accession no. EMD-7066) |
| Tang WJ | 2018 | IDE pO/pO state with FabH11-E map | http://www.ebi.ac.uk/pdbe/entry/emdb/EMD-7091 | Publicly available at the Electron Microscopy Data Bank (accession no. EMD-7091) |
| Tang WJ | 2018 | IDE with mostly insulin A chain model | http://www.ebi.ac.uk/pdbe/entry/pdb/6b3q | Publicly available at the Electron Microscopy Data Bank (accession no. 6b3q) |
| Tang WJ | 2018 | IDE with mostly insulin B chain model | http://www.ebi.ac.uk/pdbe/entry/pdb/6bfc | Publicly available at the Electron Microscopy Data Bank (accession no. 6BFC) |
| Tang WJ | 2018 | IDE O/pO state model | http://www.ebi.ac.uk/pdbe/entry/pdb/6bf8 | Publicly available at the Electron Microscopy Data Bank (accession no. 6bf8) |
| Tang WJ | 2018 | IDE O/O state model | http://www.ebi.ac.uk/pdbe/entry/pdb/6b7y | Publicly available at the Electron Microscopy Data Bank (accession no. 6b7y) |
| Tang WJ | 2018 | IDE pO/pO state model | http://www.ebi.ac.uk/pdbe/entry/pdb/6bf6 | Publicly available at the Electron Microscopy Data Bank (accession no. 6bf6) |
| Tang WJ | 2018 | IDE with insulin and FabH11-E model | http://www.ebi.ac.uk/pdbe/entry/pdb/6b70 | Publicly available at the Electron Microscopy Data Bank (accession no. 6b70) |
| Tang WJ | 2018 | IDE O/pO state with FabH11-E model | http://www.ebi.ac.uk/pdbe/entry/pdb/6bf9 | Publicly available at the Electron Microscopy Data Bank (accession no. 6bf9) |
| Tang WJ | 2018 | IDE O/O state with FabH11-E model | http://www.ebi.ac.uk/pdbe/entry/pdb/6b7z | Publicly available at the Electron Microscopy Data Bank (accession no. 6b7z) |

| Tang WJ | 2018 | IDE pO/pO state with FabH11-E model | http://www.ebi.ac.uk/pdbe/entry/pdb/6bf7 | Publicly available at the Electron Microscopy Data Bank (accession no. 6bf7) |
|---------|------|------|------|------|
| Tang WJ | 2018 | IDE with insulin and Fab1 structure | http://www.rcsb.org/pdb/search/structid-Search.do?structureId=5WOB | Publicly available at the RCSB Protein Data Bank (accession no. 5WOB) |
| Tang WJ | 2018 | IDE with amyloid $\beta$ and Fab1 structure | https://www.rcsb.org/structure/4M1C | Publicly available at the RCSB Protein Data Bank (accession no. 4M1C) |
| Tang WJ | 2018 | IDE with insulin and Fab1E structure | https://www.rcsb.org/structure/5CJO | Publicly available at the RCSB Protein Data Bank (accession no. 5CJO) |
| Tang WJ | 2018 | IDE with FabH11-E structure | https://www.rcsb.org/structure/5UOE | Publicly available at the RCSB Protein Data Bank (accession no. 5UOE) |

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
