## [Decision Letter]

Thank you for submitting your article "Ensemble cryoEM elucidates the mechanism of insulin capture and degradation by human insulin degrading enzyme" for consideration by *eLife*. Your article has been reviewed by four peer reviewers, and the evaluation has been overseen by Sriram Subramaniam as Reviewing Editor and John Kuriyan as the Senior Editor. The following individuals involved in review of your submission have agreed to reveal their identity: Marin van Heel and Cristina Paulino.

The reviewers have discussed the reviews with one another and the Reviewing Editor has drafted this decision to help you prepare a revised submission. Note that no new experiments are called for.

Summary:

In this manuscript by Zhang et al., the authors use a variety of techniques, including cryo-electron microscopy, X-ray crystallography, HDX-MS, and SAXS, to investigate the structural changes that occur in the insulin degrading enzyme (IDE) upon binding of insulin. The study presents the first structures of open and partially open IDE, and gives new insight into IDE dynamics, and how the enzyme captures, unfolds, and degrades substrates. The combination of techniques is commendable, but revisions to both text and figures are needed to both fully clarify the results, and to improve overall readability.

Essential revisions:

1) An impressive number of different techniques are used, and the results were compressed into a paper whose core is only 12 pages long. The paper should be rewritten to more fully discuss each of the structures and techniques presented, and to improve the overall flow of the manuscript.

2) In the introduction:a) The paper would gain in accessibility by expanding the main text providing more Introduction and by increasing the number of illustrations.

b) The authors indicate that IDE is composed of IDE-C and IDE-N domains for a combined mass of 110 kDa. However, in this section they should also make clear that the subunit in turn dimerizes to form a 220 kDa assembly. Figure clarity could be improved.

c) As the authors note, the IDE structure has been studied previously. A sentence or two about what state(s) these earlier structures have provided would help to orient the uninitiated reader and help illuminate why the new structures are important.

3) In the Results section:a) The argument that the binding of the Fab does not influence solution conformation is used to justify the cryo-EM structures. However, the authors also seem to imply that the fact that the X-ray structures of the Fab-IDE complex and substrate-bound state are similar implies that either antibody binding or crystal formation make it unphysiological. Please clarify.

b) The authors present the crystal structure of IDE-FabH11-E and interrogate the binding interface using HDX. This Fab is important for cryo-EM analysis, but it's not entirely clear how these X-ray and HDX results augment or weave into the overall study. Please rewrite to better meld these results into a single narrative.

c) Please include a clear description of the three X-ray structures of substrate-bound IDE. The resolution is at least as good as in the cryo-EM structure (3.3 Å, 3.5 Å, and 2.95 Å) but only Figure 7—figure supplement 1 briefly shows insulin bound in the IDE-Fab1-insulin X-ray structure. How does it appear in the better-resolved IDE-Fab1E-Insulin structure at 3.3 Å (of which there is no figure)? Additionally, please add discussion of IDE-Fab1 bound to substrate Aβ at 3.5 Å. Compare both structures in more detail and draw general conclusions about substrate binding.

d) IDE is referred to as pseudo-symmetric throughout the manuscript. Structural asymmetry arises during catalysis, but strictly speaking does the dimer not have C2 symmetry? (see additional notes for methods, below) If IDE is natively dimeric, with asymmetry emerging during catalysis, are there implications for cross-talk between the two subunits that can be derived from the new structures?

e) Subsection “SAXS analysis of IDE”: What is the degradation rate of IDE? It is stated that the insulin-induced changes might be rate-limiting, but actual numbers and comparison are missing. Also, briefly describe what Rg is.

f) Subsection “Mechanism for the processive degradation of insulin by IDE”: The authors state that the newly presented structures differ from previously published structures, but a more detailed description and visualization of how and where is needed. When do the authors suggest that the cutting takes place? In a partially closed (pC) or only in the closed state (C) of IDE? Or in both states? Please clarify.

4) In the Materials and methods section:a) For the IDE-dimer focused refinements, is the FSC resolution determination focused on an area around the IDE dimer structure by, say, a Gaussian mask cutting away most of the Fab influences (Figure 2—figure supplement 9, for example)? Please explain.

b) In this paper, all final cryo-EM resolutions are estimated "using Fourier Correlation (FSC-0.143, gold-standard)". Recently the FSC=0.143 metric was again shown to be wrong [https://www.biorxiv.org/content/early/2017/11/24/224402] but now with very convincing arguments. The authors may wish to use the necessary revision cycle to update their resolution criteria.

c) Although the problem with preferred orientation of particles in the cryo-EM data is mentioned in the main manuscript, the authors claim a resolution of 3.7 Å for the final particle-subtracted map of IDE-insulin. The authors may state that they have 3.7 Å resolution, but at the same time should indicate that it is anisotropic with a global resolution of 4.1 Å as shown in Figure 2—figure supplement 9. This would also fit better with the overall appearance of the shown density. Static figures (in addition to Video 2) should be included to show quality of map (see point (5i) for figures below).

d) Please clarify if symmetry (C2?) is imposed during reconstruction for each structure.

e) In the Materials and methods section, authors state that Relion 2.0 was used, while elsewhere authors state that Relion 2.1 was used. Which was it? Please confirm which version was used for each reconstruction.

5) In the figures:a) Overall, please increase resolution of submitted figures.

b) Introductory graphics showing location of the insulin binding site in the IDE should be improved.

c) Consistent coloring and labeling of figures should be used throughout. Additionally, please improve description of figures in the figure legends, including exactly which structure is shown (X-ray? Cryo-EM? Domain? Resolution? State?).

d) For Figure 1: The Fab label obscures fart of the Fab. Also, the title implies the IDE dimer is present, which is true for Figure 1G, while Figure 1A appears to show a monomer. Please clarify.

e) For Figure 1, Figure 4, etc.: Throughout the figures the IDE X-ray structures are shown in the monomeric form. Do they crystallize as monomers or dimers? If dimers, then are the B-factors different between subunits, and how does this impact the catalytic door analysis?

f) For Figure 5—figure supplement 2: Show B-factor for the X-ray structures?

g) In Figure 1D,E,F: "2D class averages" are depicted (also in Figure 2—figure supplement 5, Figure 2—figure supplement 6 and Figure 2—figure supplement 7). These averages look blurry while overwhelmed by low-frequency information at 5 Å resolution. The authors may wish to high-pass filter the images to emphasize the high-frequency information assumed to be present in these averages. Such high-pass filtered images would also show better what part of the "2D class averages" are stable and what parts are more flexible. In Figure 2—figure supplement 5, Figure 2—figure supplement 6 and Figure 2—figure supplement 7, provide boxes around a few representative single particles to improve figure clarity.

h) Please add higher resolution micrograph images (as an inset, perhaps?).

i) For Figure 1, Figure 4, Figure 5, Figure 2—figure supplement 1, Figure 5—figure supplement 6: Show error bars for HDX experiments?

j) The presentation of the structures in Figure 2 and beyond is without the Fab, which is misleading because the Fab was bound and has been computationally removed (?). Please clarify in figures, figure legends, and text.

k) For Figure 2—figure supplement 9, the insulin-bound IDE-FabH11-E structure is reported at 3.7 Å. Given the small size of the protein, it's warranted to present side chain densities from the cryo-EM map, as is included in Video 1 and Video 2. This could be done as representative helices and sheets.

l) Figure 6, Figure 8—figure supplement 2: These figures are problematic because they don't give the reader a true impression of the ligand density. To address this, at some point in the manuscript the authors could establish the ligand density strength by showing it at the same contour as the surrounding protein, rather than masking out just the ligand. Additional, the masking approach used to display the ligand density should be described.

m) Supplementary file 2: With regards to pixel size, should magnification be 45.454 rather than 22.500? Some particle numbers in Supplementary file 2 and Supplementary file 3 don't match.

n) Video legends: For Video 1, the wrong resolutions are indicated. For Video 2, legend is missing. For Video 3, legend is indicated as legend for Video 2.

---

## [Author Response]

Summary:In this manuscript by Zhang et al., the authors use a variety of techniques, including cryo-electron microscopy, X-ray crystallography, HDX-MS, and SAXS, to investigate the structural changes that occur in the insulin degrading enzyme (IDE) upon binding of insulin. The study presents the first structures of open and partially open IDE, and gives new insight into IDE dynamics, and how the enzyme captures, unfolds, and degrades substrates. The combination of techniques is commendable, but revisions to both text and figures are needed to both fully clarify the results, and to improve overall readability.Essential revisions:1) An impressive number of different techniques are used, and the results were compressed into a paper whose core is only 12 pages long. The paper should be rewritten to more fully discuss each of the structures and techniques presented, and to improve the overall flow of the manuscript.

We have substantially revised the manuscript.

2) In the introduction:a) The paper would gain in accessibility by expanding the main text providing more Introduction and by increasing the number of illustrations.

We have added an introductory figure and additional text to elaborate the background and critical unsolved issues. We have also substantially revised and expanded the illustrations.

b) The authors indicate that IDE is composed of IDE-C and IDE-N domains for a combined mass of 110 kDa. However, in this section they should also make clear that the subunit in turn dimerizes to form a 220 kDa assembly. Figure clarity could be improved.

Where it is appropriate and does not make figure too complicated, we have revised figures to illustrate the fact that IDE is a dimer.

c) As the authors note, the IDE structure has been studied previously. A sentence or two about what state(s) these earlier structures have provided would help to orient the uninitiated reader and help illuminate why the new structures are important.

We have added the introduction figure and expand the Introduction to provide the readers the necessary background in the previous structural analysis and critical unresolved structures.

3) In the Results section:a) The argument that the binding of the Fab does not influence solution conformation is used to justify the cryo-EM structures. However, the authors also seem to imply that the fact that the X-ray structures of the Fab-IDE complex and substrate-bound state are similar implies that either antibody binding or crystal formation make it unphysiological. Please clarify.

It is well accepted that crystallography is the “prisoner of crystal lattice”. We hypothesize that the crystal lattice has placed constraints so that IDE prefers to be in the closed state, resulting that alcrystal lattice has placed constraints so that IDE prefers to be in the closed state, resulting that all crystallization conditions that we and others have found trap IDE in the closed state regardless whether the substrates are present or not. We do not imply that the crystallization condition is nonphysiological. We add a short comment in the result section. This is also why we applied cryoEM for our studies.

b) The authors present the crystal structure of IDE-FabH11-E and interrogate the binding interface using HDX. This Fab is important for cryo-EM analysis, but it's not entirely clear how these X-ray and HDX results augment or weave into the overall study. Please rewrite to better meld these results into a single narrative.

We modify the Results section to better incorporate HDX-MS data into this section.

c) Please include a clear description of the three X-ray structures of substrate-bound IDE. The resolution is at least as good as in the cryo-EM structure (3.3 Å, 3.5 Å, and 2.95 Å) but only Figure 7—figure supplement 1 briefly shows insulin bound in the IDE-Fab1-insulin X-ray structure. How does it appear in the better-resolved IDE-Fab1E-Insulin structure at 3.3 Å (of which there is no figure)? Additionally, please add discussion of IDE-Fab1 bound to substrate Aβ at 3.5 Å. Compare both structures in more detail and draw general conclusions about substrate binding.

In our previous version, we have compared the crystal structures with cryoEM structures throughout the paper. We realize that we have not made it very clear in our result section. We now clearly label all of crystal structures and cryoEM structures in the figures to avoid the perception that we did not discuss our crystal structure fully.

d) IDE is referred to as pseudo-symmetric throughout the manuscript. Structural asymmetry arises during catalysis, but strictly speaking does the dimer not have C2 symmetry? (see additional notes for methods, below) If IDE is natively dimeric, with asymmetry emerging during catalysis, are there implications for cross-talk between the two subunits that can be derived from the new structures?

Accumulating evidences support dimer-mediated IDE allostery. We have added substantial discussion about the potential impact on the structural basis of allosteric regulation of IDE based on our structural analysis.

e) Subsection “SAXS analysis of IDE”: What is the degradation rate of IDE? It is stated that the insulin-induced changes might be rate-limiting, but actual numbers and comparison are missing. Also, briefly describe what Rg is.

The rate of degradation of IDE is substrate dependent. We have stated that the rate of insulin degradation by IDE is ~2 per second in the previous version. To highlight that, we state such value whenever it is appropriate. We also add the description of Rg as suggested.

f) Subsection “Mechanism for the processive degradation of insulin by IDE”: The authors state that the newly presented structures differ from previously published structures, but a more detailed description and visualization of how and where is needed. When do the authors suggest that the cutting takes place? In a partially closed (pC) or only in the closed state (C) of IDE? Or in both states? Please clarify.

We have done the detailed comparison using several measurements shown in Figure 3. We envision that the catalysis occurs in both pC and C IDE states. We add some discussion in the Results section and Discussion section.

4) In the Materials and methods section:a) For the IDE-dimer focused refinements, is the FSC resolution determination focused on an area around the IDE dimer structure by, say, a Gaussian mask cutting away most of the Fab influences (Figure 2—figure supplement 9, for example)? Please explain.

Indeed yes, text has now been added to the Materials and methods section to clarify this point.

b) In this paper, all final cryo-EM resolutions are estimated "using Fourier Correlation (FSC-0.143, gold-standard)". Recently the FSC=0.143 metric was again shown to be wrong [https://www.biorxiv.org/content/early/2017/11/24/224402] but now with very convincing arguments. The authors may wish to use the necessary revision cycle to update their resolution criteria.

While we accept the value of the reviewer’s work in this area we prefer to leave the resolution criteria as they are until the field as a whole decides to adopt this new criterion. We do not believe that this paper, focused almost entirely on the biological results, is the appropriate forum to enter into a discussion on this point.

c) Although the problem with preferred orientation of particles in the cryo-EM data is mentioned in the main manuscript, the authors claim a resolution of 3.7 Å for the final particle-subtracted map of IDE-insulin. The authors may state that they have 3.7 Å resolution, but at the same time should indicate that it is anisotropic with a global resolution of 4.1 Å as shown in Figure 2—figure supplement 9. This would also fit better with the overall appearance of the shown density.

The 3.7 Å resolution was estimated by RELION. The anisotropy was analyzed by 3DFSC (Tan et al., 2017). The angular distribution of IDE-insulin and the sphericity of the structure of 0.87 indeed do indicate resolution anisotropy. We have now added a comment to this effect in the Materials and methods section and we have added an extra row in Supplementary file 3 to show the global resolution.

Static figures (in addition to Video 2) should be included to show quality of map (see point (5i) for figures below).

We have added an extra extended data in Figure 3—figure supplement 1 to show both the overall density map and representative densities of α helixes.

d) Please clarify if symmetry (C2?) is imposed during reconstruction for each structure.

C1 symmetry was used in all cryoEM structures and this is now stated in the methods. No noncrystallographic symmetry related to IDE dimer was applied to crystal structures.

e) In the Materials and methods section, authors state that Relion 2.0 was used, while elsewhere authors state that Relion 2.1 was used. Which was it? Please confirm which version was used for each reconstruction.

We thank the reviewer for spotting this error. We used Relion2.1 throughout refinement, and we have updated the Materials and methods section to reflect this.

5) In the figures:a) Overall, please increase resolution of submitted figures.

Modified as suggested.

b) Introductory graphics showing location of the insulin binding site in the IDE should be improved.

We have modified introduction as suggested and added new introductory figure.

c) Consistent coloring and labeling of figures should be used throughout. Additionally, please improve description of figures in the figure legends, including exactly which structure is shown (X-ray? Cryo-EM? Domain? Resolution? State?).

We have followed the recommendation to modify the main and supplementary figures.

d) For Figure 1: The Fab label obscures fart of the Fab. Also, the title implies the IDE dimer is present, which is true for Figure 1G, while Figure 1A appears to show a monomer. Please clarify.

We have modified the figure accordingly.

e) For Figure 1, Figure 4, etc.: Throughout the figures the IDE X-ray structures are shown in the monomeric form. Do they crystallize as monomers or dimers? If dimers, then are the B-factors different between subunits, and how does this impact the catalytic door analysis?

All crystallographic structures of IDE are dimer. We now modified the figures accordingly. We also add the new figure panels.

f) For Figure 5—figure supplement 2: Show B-factor for the X-ray structures?

We added the B factor for the x-ray structures as recommended.

g) In Figure 1D,E,F: "2D class averages" are depicted (also in Figure 2—figure supplement 5, Figure 2—figure supplement 6 and Figure 2—figure supplement 7). These averages look blurry while overwhelmed by low-frequency information at 5 Å resolution. The authors may wish to high-pass filter the images to emphasize the high-frequency information assumed to be present in these averages. Such high-pass filtered images would also show better what part of the "2D class averages" are stable and what parts are more flexible.

In Figure 1 and Figure 2—figure supplement 5, Figure 2—figure supplement 6 and Figure 2—figure supplement 7, 2D class average are now shown calculated in Relion2.1 using the option to ignore the CTFs until the first peak, which effectively damps down the low frequency components and sharpens up high frequency information.

In Figure 2—figure supplement 5, Figure 2—figure supplement 6 and Figure 2—figure supplement 7, provide boxes around a few representative single particles to improve figure clarity.

We provide 2 raw micrographs each of untilted apoIDE, tilted apoIDE and insulin bound IDE Figure 2—figure supplement 5, Figure 2—figure supplement 6 and Figure 2—figure supplement 7. Representative particles have been circled in red and higher magnification views of these particles are shown as insets.

h) Please add higher resolution micrograph images (as an inset, perhaps?).

We have incorporated high resolution micrograph images in Figure 2—figure supplement 5, Figure 2—figure supplement 6 and Figure 2—figure supplement 7.

i) For Figure 1, Figure 4, Figure 5, Figure 2—figure supplement 1, Figure 5—figure supplement 6: Show error bars for HDX experiments?

We add standard deviation error bar to the WT IDE data which was determined by three individual experiments. We also modified to method section to explain the reproducibility of experimental data and the statistical significance of changes.

j) The presentation of the structures in Figure 2 and beyond is without the Fab, which is misleading because the Fab was bound and has been computationally removed (?). Please clarify in figures, figure legends, and text.

We have clarified this accordingly.

k) For Figure 2—figure supplement 9, the insulin-bound IDE-FabH11-E structure is reported at 3.7 Å. Given the small size of the protein, it's warranted to present side chain densities from the cryo-EM map, as is included in Video 1 and Video 2. This could be done as representative helices and sheets.

New Figure 7—figure supplement 1 is added.

l) Figure 6, Figure 8—figure supplement 2: These figures are problematic because they don't give the reader a true impression of the ligand density. To address this, at some point in the manuscript the authors could establish the ligand density strength by showing it at the same contour as the surrounding protein, rather than masking out just the ligand. Additional, the masking approach used to display the ligand density should be described.

We have added the additional panel for the comparison.

m) Supplementary file 2: With regards to pixel size, should magnification be 45.454 rather than 22.500? Some particle numbers in Supplementary file 2 and Supplementary file 3 don't match.

Yes, the magnification should be 45,454X. The magnification 22,500 in Supplementary file 2 refers to the nominal magnification as set by the microscope. This has been corrected.

n) Video legends: For Video 1, the wrong resolutions are indicated. For Video 2, legend is missing. For Video 3, legend is indicated as legend for Video 2.

Modified accordingly.